# IL-2 delivery to CD8$^+$ T cells during infection requires MRTF/SRF-dependent gene expression and cytoskeletal dynamics

Diane Maurice [1,4], Patrick Costello [1], Jessica Diring [1], Francesco Gualdrini [1,2], Bruno Frederico [3,5] & Richard Treisman [1] ✉

Paracrine IL-2 signalling drives the CD8 + T cell expansion and differentiation that allow protection against viral infections, but the underlying molecular events are incompletely understood. Here we show that the transcription factor SRF, a master regulator of cytoskeletal gene expression, is required for effective IL-2 signalling during *L. monocytogenes* infection. Acting cell-autonomously with its actin-regulated cofactors MRTF-A and MRTF-B, SRF is dispensable for initial TCR-mediated CD8$^+$ T cell proliferation, but is required for sustained IL-2 dependent CD8$^+$ effector T cell expansion, and persistence of memory cells. Following TCR activation, *Mrtfab*-null CD8$^+$ T cells produce IL-2 normally, but homotypic clustering is impaired both in vitro and in vivo. Expression of cytoskeletal structural and regulatory genes, most notably actins, is defective in *Mrtfab*-null CD8$^+$ T cells. Activation-induced cell clustering in vitro requires F-actin assembly, and *Mrtfab*-null cell clusters are small, contain less F-actin, and defective in IL-2 retention. Clustering of *Mrtfab*-null cells can be partially restored by exogenous actin expression. IL-2 mediated CD8$^+$ T cell proliferation during infection thus depends on the control of cytoskeletal dynamics and actin gene expression by MRTF-SRF signalling.

During infection, regulation of CD8$^+$ T-cell migration, proliferation and differentiation plays critical roles in successful pathogen clearance and establishment of protective immunity[1–3]. Upon interaction of previously unstimulated T cells with antigen-presenting cells (APCs), the formation of the immune synapse facilitates TCR activation by peptide-MHC complexes and co-stimulatory adhesive signalling[4,5]. Activated CD8$^+$ T cells differentiate into highly proliferative short-lived effector cells (SLECs), which mediate pathogen clearance and undergo subsequent apoptosis, and a smaller population of memory precursor cells (MPEC), which give rise to self-renewing memory cells[6–8]. Strong TCR signals favour SLEC differentiation[8,9], while inactivation of mTOR favours MPEC formation and memory cell generation[10].

While the initial proliferative response of CD8$^+$ T cells is TCR-dependent, their subsequent proliferation and differentiation is modulated by IL-2 and other cytokines[11–14] (see refs. 15,16 for review). The strength of IL-2 signalling controls the balance between effector and memory cell formation[17–19], and the cytokine signal transducer STAT5 is essential for effector cell expansion[20,21]. Both IL-2 and the transcription factor IRF4 are dispensable for early CD8$^+$ T-cell proliferation but necessary for sustained expansion and SLEC proliferation[11,18,22–24]. Effective cytokine delivery and activity is dependent on hetero- and homotypic T-cell interactions[14,25–27]. In keeping with this, SLEC expansion is dependent on the T-cell integrin LFA-1 (αLβ2; CD11a/CD18)[26,28,29], its ligand ICAM[26], and its effector kinase PYK2[28].

[1]Signalling and transcription Laboratory, Francis Crick Institute, 1 Midland Road, London NW1 1AT, UK. [2]European Institute of Oncology (IEO), Instituto di Ricovero e Cura a Carattere Scientifico (IRCCS), Milan 20139, Italy. [3]Immunobiology Laboratory, Francis Crick Institute, 1 Midland Road, London NW1 1AT, UK. [4]Present address: Autoimmunity Laboratory, Francis Crick Institute, 1 Midland Road, London NW1 1AT, UK. [5]Present address: Early Oncology, R&D AstraZeneca, Cambridge, UK. ✉e-mail: Richard.Treisman@crick.ac.uk

The SRF transcription factor network controls receptor-regulated and cytoskeletal gene expression[30–32] through interaction with two families of signal-responsive regulatory co-factors[33,34]. The ERK-regulated ternary complex factors (TCFs) control classical immediate-early genes, and are essential for thymocyte positive selection[35–38]. The Myocardin-Related Transcription Factors (MRTFs), which act as sensors of cellular G-actin concentration, control dozens of cytoskeletal structural and regulatory genes[39–42]. MRTF-SRF signalling is essential for effective cytoskeletal dynamics[32,34,40,43], and at least in some contexts this appears to reflect its role in homoeostatic control of actin gene expression[43]. MRTF-SRF signalling is essential for seeding of HSC/P in the bone marrow[44] and for thymocyte migration (PC, DM and RT, in preparation).

Here we investigate the role of the SRF network in the proliferative response of CD8+ T cells during *L. monocytogenes* infection. Although the initial proliferative response to TCR activation is intact, cells lacking SRF or the MRTFs are unable to maintain sustained IL-2 dependent expansion of activated CD8+ T cells. IL-2 is produced normally and cells retain the ability to respond to exogenous IL-2. *Mrtfab*-null CD8+ T cells exhibit defective homotypic T-cell clustering in vitro and in vivo and reduced basal and IL-2 induced transcription of MRTF-SRF cytoskeletal target genes, including the cytoplasmic actins. *Mrtfab*-null clusters are smaller, with reduced F-actin content, and ineffectively retain endogenous IL-2. Strikingly clustering, which is dependent on F-actin assembly, can be partially restored by exogenous actin gene expression in *Mrtfab*-null cells. Our findings demonstrate that MRTF/SRF-dependent cytoskeletal dynamics are essential for the homotypic CD8+ T-cell interactions that underlie IL-2 induced CD8+ T-cell proliferation during infection.

## Results

### SRF is essential for peripheral T-cell expansion upon *Listeria* infection

To study CD8+ T-cell differentiation during infection, we used the well-characterized *L. monocytogenes*-OVA (LM-OVA) infection model[45,46]. SRF is essential for T-cell development, so to understand its role in peripheral T-cell function, we exploited a tamoxifen-inducible conditional *Srf* allele coupled with bone marrow reconstitution. Bone marrow from *Srf*f/f or *Srf*+/+ (WT) animals ubiquitously expressing a tamoxifen-regulated Cre derivative and the OVA-specific OT-I TCR (OT-I *Srf*f/f TamCre; CD45.2, and OT-I WT TamCre; CD45.1) was used to reconstitute RAG-2−/− mice. Six weeks later, mice were fed with tamoxifen for 15 days to inactivate *Srf* and allow SRF protein depletion in hematopoietic tissues (Fig. 1A, B, S1A). The resulting OT-I WT TamCre and OT-I *Srf*−/− TamCre CD8 T cells had the properties of naive, unstimulated cells, as judged by their profiles of the cell surface markers CD62L, CD44, CD5, and CCR7 (Fig. S1B), and their failure to produce cytokines upon OVA peptide stimulation in vitro (Fig. S1C).

To compare the responses of wildtype and *Srf*-null CD8+ T cells to LM-OVA infection in the same environment we used an adoptive co-transfer strategy (Fig. 1A). OT-I WT and OT-I *Srf*−/− CD8+ T cells were combined at 1:1 ratio, and 10,000 cells co-transferred into congenic C57BL/6 CD45.1/2 recipients, which were challenged 24 h later with a priming dose of LM-OVA. Expansion of OT-I *Srf*−/− effector CD8+ T cells was greatly reduced in blood and spleen, and maximal at day 6 (Fig. 1C). At day 8, the peak of wildtype CD8+ T-cell expansion, both OT-I WT and OT-I *Srf*−/− CD8+ T cells expressed CD44 similarly, indicating that activation was intact. Both the adhesion molecule L-selectin/CD62L, and IL-7Rα, markers of memory differentiation, were initially downregulated normally in OT-I *Srf*−/− CD8+ T cells, but by day 8 an enhanced proportion had re-acquired expression (Fig. S1D). Consistent with this, the OT-I *Srf*−/− CD8+ population displayed a dramatic reduction in short-lived effector cells, and a decreased SLEC:MPEC ratio[6,7](Fig. 1D). Upon ex vivo stimulation with OT-I specific OVA peptide SIINFEKL, OT-I *Srf*−/− T cells produced increased amounts of TNF-α

and IFN-γ compared with OT-I WT cells, but granzyme B expression was slightly decreased, suggesting their differentiation was affected (Fig. 1E). By day 32 post-infection, when fully differentiated memory CD8+ T cells should be present, OT-I *Srf*−/− CD8+ T-cell numbers had declined below the limit of detection (Fig. 1C). During LM-OVA infection, SRF thus plays an essential and cell-autonomous role in robust CD8+ T-cell expansion and in generation and persistence of memory cells.

### SRF is required for CD8+ T memory cell expansion

To generate memory CD8+ T cells lacking SRF, we used the R26TamCre system to inactivate *Srf* once the response to initial infection was under way. OT-I WT and OT-I Srff/f cells were adoptively transferred to wild-type mice at 1:1 ratio, and the recipients treated with tamoxifen 4 days after LM-OVA infection (Fig. 2A). In this setting, SRF was effectively depleted by day 6 post-infection (Fig. 2B). Expansion of these OT-I "*Srf*−/−(post)" cells in blood was maximal at day 8: their SLEC-MPEC ratio was substantially unaltered (Fig. S2A), and they produced IFN-γ at comparable levels to OT-I WT cells upon activation with OVA peptide (Fig. S2B).

OT-I *Srf*−/−(post) cells contracted with similar kinetics to OT-I WT cells, and putative *Srf*-null memory cells were readily detectable at day 45 (Figs. 2C and S2C, D). These expressed elevated levels of the central memory T-cell markers CD62L, CCR7 and IL-2Rβ, suggesting that late inactivation of *Srf* might enhance acquisition of central memory characteristics (Fig. S2C). Memory OT-I WT and *Srf*−/−(post) cells, defined by CD62L, CCR7 and IL-2Rβ expression, were purified and co-transferred at 1:1 ratio into new recipients (Fig. 2D). The transferred *Srf*−/−(post) memory cells exhibited greatly reduced expansion upon LM-OVA challenge: at day 6 after infection, when the wildtype recall response reached its peak, the OT-I *Srf*−/−(post) numbers were only 20% that of wildtype (Fig. 2E). Moreover, the re-transferred OT-I *Srf*−/−(post) CD8+ memory cell population did not maintain expression of IL-2Rα at day 6 following infection (Fig. 2F). Thus, *Srf* is essential not only for the primary CD8+ T-cell expansion but for reactivation and expansion of CD8+ T memory cells in response to infectious challenge.

### Requirement for SRF reflects the activity of its MRTF co-factors

We next used the co-transfer approach to investigate the contributions of the two families of signal-regulated SRF co-factors, the TCFs and the MRTFs[33], to the CD8+ T-cell response during LM-OVA infection. SAP-1, encoded by *Elk4*, is the most abundant TCF in T cells and is required with SRF for T-cell development[35–38] (Fig. 3A). Inactivation of *Elk4* affected neither the CD8+ T-cell proliferative response (Fig. 3B) nor the balance between SLECs and MPECs following infection (Fig. 3C).

To investigate the role of the MRTFs, we used a similar deletion strategy to that used for SRF, exploiting the viability of *Mrtfa*-null mice in conjunction with the conditional *Mrtfb*fl/fl allele[47]. RAG-2 deficient hosts were reconstituted with OT-I WT TamCre CD45.1 or OT-I *Mrtfa*−/− *Mrtfb*fl/fl TamCre CD45.2 bone marrow, treated with tamoxifen to inactivate *Mrtfb*, and subsequently co-transferred with OT-I WT cells to new hosts for infection analysis (Fig. 3D, E). The transferred OT-I *Mrtfab*−/− CD8+ T cells were naive, as assessed by the expression of CD62L and the activation marker CD44 (Fig. S3A). As with Srf inactivation, by day 8 an enhanced proportion of *Mrtfab*-null cells had re-acquired CD62L and IL-7Rα expression (Fig. S3B). Upon LM-OVA infection, expansion of OT-I *Mrtfab*−/− CD8+ T cells was substantially attenuated (Fig. 3F), and the SLEC/MPEC ratio was markedly reduced, with greatly impaired SLEC accumulation (Fig. 3G).

Taken together, these results show that the requirement for SRF in CD8+ T-cell expansion following LM-OVA infection reflects the activity of its MRTF co-factors. Even though the TCF-SRF arm of the SRF network remains intact in *Mrtfab*−/− cells, it cannot compensate for the loss of the MRTFs. Moreover, the similarity between the *Srf*−/− and *Mrtfab*−/− proliferative phenotypes suggests that TCF-SRF signalling is

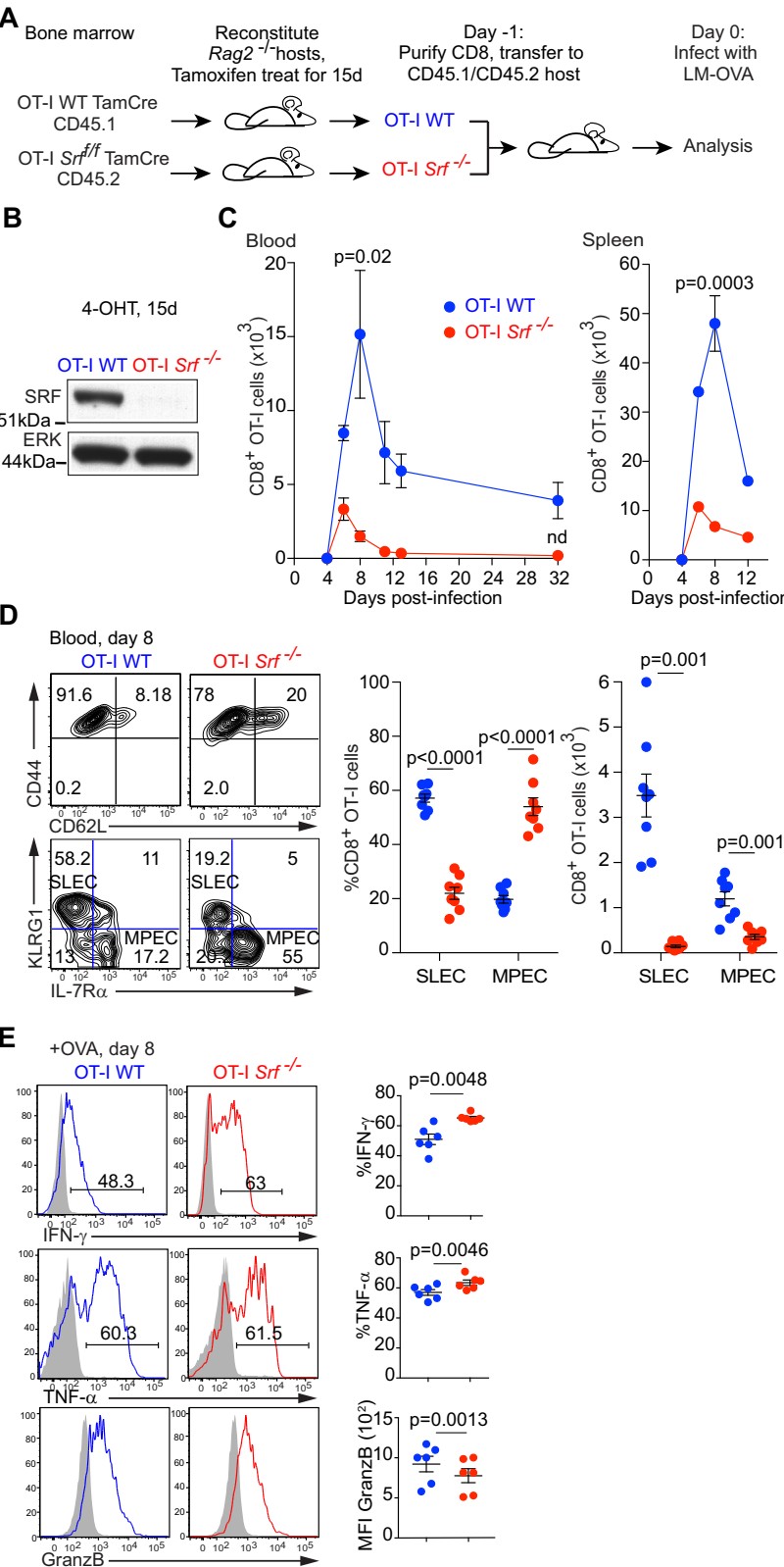

not required for the residual proliferative response seen in *Mrtfab*−/− cells (see Discussion).

## Sustained proliferation requires MRTF-SRF signalling

To gain insight into the nature of the expansion defect in *Srf*-null T cells, we examined the initial stages of infection in the spleen, the primary site of *L. monocytogenes* infection. To enable cell tracking during the early stages of infection, we used larger numbers of OT-I WT and OT-I *Srf*−/− cells for adoptive transfer (1–2 × 10⁶ each genotype). Following LM-OVA infection, both OT-I WT and OT-I-*Srf*−/− cell populations increased similarly until day 3 post infection, but OT-I *Srf*−/− cells exhibited significantly reduced accumulation at day 4 (Fig. 4A). A similar result was obtained with OT-I *Mrtfab*−/− cells (Fig. 4B), which localised to the T-cell zones of the spleen, and upregulated the IL-2

**Fig. 1 | Srf is essential for the CD8+ T-cell response to Listeria infection.**
**A** Experimental protocol. Irradiated *Rag2*−/− mice were reconstituted with bone marrow from OT-I WT TamCre or OT-I *Srf* f/f TamCre mice and the resulting chimeras fed with tamoxifen for 15 days. Purified OT-I WT (CD45.1) and OT-I *Srf*−/− (CD45.2) naive CD8+ T cells were then co-transferred 1:1 into CD45.1/CD45.2 recipients, and infected with rLM-OVA the following day. **B** Immunoblot analysis of SRF protein in *Rag2*−/− reconstituted mice following 15 days tamoxifen treatment reveals complete *Srf* inactivation in OT-I *Srf*−/− cells as also shown in Fig, S1A. **C** Flow cytometry analysis following adoptive co-transfer of 5000 each OT-I WT and OT-I *Srf*−/− CD8+T cells upon rLM-OVA infection. Data are mean numbers ± SEM in blood (20 µl) or spleen, *n* = 3 mice per time point, and are representative of ≥3 independent experiments; nd, not detectable. Statistical significance, paired two-tailed t test. **D** Cell surface expression of activation markers CD44 and CD62L, and MPEC/SLEC markers KLRG1 and IL-7Rα on OT-I WT TamCre (CD45.1 gated) and OT-I *Srf*−/− TamCre (CD45.2 gated) at day 8 post-infection in blood. Right panel, Proportions and numbers of SLEC (KLRG1hi, IL-7Rαlow) and MPEC (KLRG1low, IL-7Rαhi) are shown with *p* values determined by paired two-tailed t test. Data are means ± SEM; datapoints are individual mice, *n* = 6 for OT-I WT and *n* = 8 for OT-I *Srf*−/− TamCre, and are representative of ≥ 3 independent experiments. **E** CD8+ T cells were isolated 8 days post-infection, activated with 10 nM SIINFEKL OVA peptide for 5 h, and stained for IFN-γ, TNF-α, and GranzB. Grey, isotype control; black, experimental sample. Data are mean values ± SEM; datapoints represent individual mice and are representative of 2 independent experiments. Statistical significance, paired two-tailed t test. Source data are provided as a Source Data file.

receptor normally following LM-OVA infection (Fig. S4A, B). These expansion defects likely reflect decreased proliferation rather than increased cell death, since BrdU incorporation by OT-I *Srf*−/− cells subsided after day 3 post-infection, while the amounts of active caspase-3 were similar to those in wildtype cells (Fig. 4C). These observations suggest that initial TCR activation remains intact in OT-I *Srf*−/− and OT-I *Mrtfab*−/− cells; indeed, naive cells proliferated if anything more effectively than wildtype following activation in vitro by plate-bound anti-CD3/CD28 or OVA peptides (Figs. 4D and S4C–E; see Discussion). IL-2 cytokine production and secretion occurred normally following TCR activation of OT-I *Srf*−/− and OT-I *Mrtfab*−/− cells, reflecting induction of IL-2 gene transcription (Fig. 4E, F). Secretion of TNFα and IFNγ also occured normally when CD8+ T cells were activated by PDBu/ionomycin (Fig. S4F). Taken together, these results show that MRTF-SRF signalling is required for effective CD8+ T-cell proliferation at a step subsequent to initial T-cell activation.

### CD8+ T-cell expansion failure is accompanied by IL-2 signalling deficits

The above results establish that following LM-OVA infection CD8+ T cells are activated and initiate proliferation, but expansion is not sustained in the absence of MRTF or SRF. We therefore investigated cell signalling during the transition to MRTF-SRF-dependent cell proliferation at days 3-4 post-infection. In both *Mrtfab*−/− and *Srf*−/− cells expression of IL-2 Rα, the high-affinity receptor for IL-2, which is induced both by TCR stimulation and by IL-2[13,17,19] began to decline by day 3, before the onset of the expansion defect (Figs. 5A, B; and S5A); expression of IRF4, which is induced by TCR activation but also responsive to IL-2[22,48], behaved in a similar manner (Figs. 5B and S5A).

In OT-I WT cells, activating phosphorylation of Akt T308 and phosphorylation of S6 S240/244 were readily detectable at day 3 (Fig. 5B); phosphorylation of S6 was sensitive to the mTORC1 inhibitor rapamycin, consistent with the involvement of PI3K-Akt-mTOR signalling (Fig. S5B). In contrast, both OT-I *Srf*−/− and OT-I *Mrtfab*−/− CD8+ T cells exhibited reduced Akt T308 and S6 S240/244 and S235/236 phosphorylation, and the latter was not reduced by rapamycin treatment (Figs. S5A and S5B). Since IL-2 induces S6 phosphorylation via PI3K-Akt-mTOR signalling[49], these results indicate that IL-2 signalling is defective in *Mrtfab*- and *Srf*-null CD8 T cells. Consistent with this view, STAT5a/b Y694 phosphorylation, a specific reporter of cytokine signalling (reviewed by refs. 15,50), was significantly reduced in OT-I *Mrtfab*−/− CD8+ T cells (Fig. 5B). These data suggest the failure to sustain CD8+ T-cell expansion in *Mrtfab*- and *Srf*-null cells reflects a defect in IL-2 signalling.

### *Srf*−/− and *Mrtfab*−/− CD8+ T cells remain responsive to IL-2

We next tested whether the defect in IL-2 signalling lies at the level of signal receipt or response. *Mrtfab*- or *Srf*-null splenic T cells were harvested 3 days post-infection, when proliferation of OT-I WT and OT-I *Mrtfab*−/− cells was comparable, and cultured for 72 h either alone, with IL-2, or with IL-12, which allows maintenance of IL-2Rα

expression[51]. In the absence of added cytokine, OT-I WT cells expanded more than OT-I *Mrtfab*−/− cells, and IL-2Rα expression declined to comparable levels on both populations (Fig. 6A). Wildtype and *Mrtfab*−/− cells responded comparably to titration of exogenous IL-2, which substantially increased both IL-2 Rα expression and proliferation to comparable extents (Figs. 6A, and S6A); in contrast, IL-12 restored IL-2Rα expression, but did not enhance proliferation (Fig. 6A). Thus, MRTF-SRF signalling is not required for CD8+ cells to respond to IL-2 in vitro, or for IL-2 Rα expression per se. Consistent with this, following activation and culture in IL-2, OT-I WT and OT-I *Srf*−/− cells purified from uninfected mice exhibited comparable abilities to kill OVA-pulsed EL4 target cells in vitro, as assessed by intracellular active caspase-3 staining (Fig. S6B).

Next, we potentiated IL-2 signalling in vivo using IL-2/S4B6 antibody complexes, which prolong the half-life of IL-2, and signal through IL-2Rβ (CD122)[52]. IL-2/S4B6 injection from day 1 to day 3 post-infection potentiated IL-2Rα expression, ERK and AKT activation, and S6 phosphorylation in both OT-I WT and OT-I *Mrtfab*−/− cells at day 4, and restored activation and proliferation of OT-I *Mrtfab*−/− cells to levels comparable to wildtype (Fig. 6B). Consistent with this, injection of IL-2/S4B6 complexes at days 3–5 post-infection substantially boosted the expansion of both OT-I WT and OT-I *Srf*−/− cells (Fig. 6C). Thus cells deficient in MRTF-SRF signalling remain sensitive to exogenously supplied IL-2.

### MRTF-dependent homotypic clustering is required for IL-2 signalling following TCR activation

The experiments in the preceding sections show that although *Mrtfab*-null and *Srf*-null CD8+ T cells retain the ability to produce IL-2 in response to TCR activation, and to respond to exogenous IL-2, they nevertheless exhibit defective ability to receive endogenous IL-2 signals. As a result, they fail to maintain IL-2Rα expression and proliferative expansion after the priming phase. To investigate the basis of this defect, we first used an in vitro approach to assess how endogenous IL-2 signalling contributes to IL-2Rα expression following TCR activation. At 24 h following TCR activation, IL-2 blockade reduced IL-2Rα expression and abolished STAT5 Y694 phosphorylation, indicating that even at this early time IL-2 signalling is active (Fig. 7A, B). IL-2 blockade also inhibited the sustained expression of IRF4 at late times during continuous TCR activation (Fig. S7A). Although IL-2Rα expression remained essentially intact, OT-I *Mrtfab*−/− cells exhibited decreased STAT5 Y694 phosphorylation, indicating a defect in IL-2 signal transduction (Fig. 7A, B). Thus, both the TCR and endogenously-produced IL-2 contribute to IL-2Rα expression, even at early times of activation.

To investigate signalling by endogenous IL-2 in the absence of continuing TCR activation, we activated cells for 24 h and then transferred them to a new plate without anti-CD3/CD28. IL-2Rα expression in OT-I WT cells declined slowly following transfer, but declined much more rapidly in OT-I *Mrtfab*−/− cells; upon IL-2 blockade the kinetics of IL-2Rα decline were similar in both genotypes, presumably reflecting declining TCR activity (Fig. 7A). Signalling by the IL-2 receptor, as

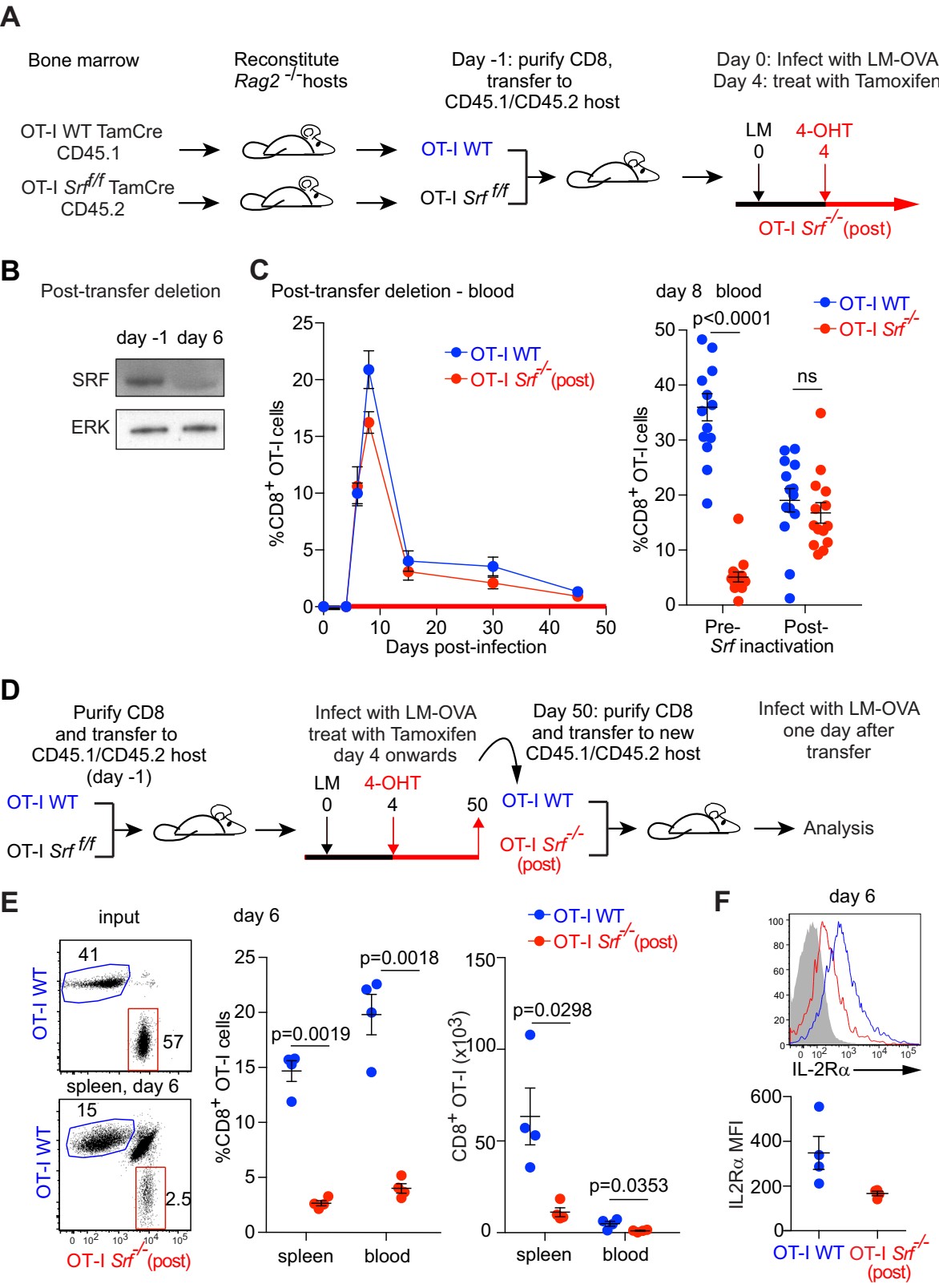

measured by STAT5 Y694 phosphorylation, initially increased upon transfer in OT-I WT, but not OT-I *Mrtfab*[−/−] cells, before decreasing at later times, and the increase was dependent on IL-2 (Fig. 7B). Thus in this setting, continued expression of IL-2Rα is dependent both on the MRTFs and endogenous IL-2. The rapid down-regulation of IL-2Rα in *Mrtfab*-null cells upon cessation of TCR activation is therefore a consequence of defective IL-2 signalling rather than its cause.

Homotypic clustering plays an important role in T-cell responses to cytokines[25,26], so we examined cell interactions in the transfer assay. Upon transfer to culture without anti-CD3/CD28, activated OT-I WT cells rapidly came together into large clusters, whose formation was substantially inhibited upon blockade of IL-2 or CD11a (Fig. 7C, Fig. S7B). In contrast, cluster formation by activated OT-I *Mrtfab*[−/−] cells was greatly reduced, even though cell surface expression of LFA-1 and its

**Fig. 2 | Srf-null memory cells cannot mount an effective recall response.**
**A** Protocol for post-infection inactivation of *Srf*. For post-infection, WT OT-I
TamCre (CD45.1) and OT-I *Srf*^f/f TamCre (CD45.2) CD8⁺ T cells were co-injected at a
ratio 1:1 into CD45.1/CD45.2 hosts, which were infected with rLM-OVA the next day.
OT-I *Srf*⁻/⁻(post): four days after infection, animals were fed tamoxifen to induce *Srf*
inactivation. **B** Immunoblotting analysis of *Srf* following tamoxifen administration. **C** Expansion of WT OT-I TamCre and OT-I *Srf*⁻/⁻(post) CD8⁺ T cells in blood. Data
show mean values ± SEM, 3 mice per time point. Representative of two independent
experiments. Right, percentages of OT-I WT and OT-I *Srf*⁻/⁻ CD8⁺ cells at day 8 after
infection in the blood of animals. OT-I *Srf*⁻/⁻(post): four days after infection, animals
were fed tamoxifen to induce *Srf* inactivation. OT-I *Srf*⁻/⁻(pre): animals transferred
with *Srf*-null cells as in Fig. 1A. Data from 3 independent experiments, one animal
per data point and paired two-tailed t test. **D** A schematic of OT-I WT (CD45.1) and

OT-I *Srf*⁻/⁻(post) (CD45.2) memory cell populations purified at day 50 after primary
LM-OVA infection from spleen of animals treated with the post-infection deletion
regime as in Fig. 2A and transferred at 1:1 ratio to a secondary CD45.1/CD45.2 host,
which was subjected to infection the following day. **E** Expansion profiles, percentages and absolute cell counts of OT-I WT and OT-I *Srf*⁻/⁻(post) memory cells are
compared 6 days after infection. Left, representative plot of input and expanded
CD8 + T cells. Right, data from 4 individual mice after infection at day 6. Dot plot
represent individual mice, Mean values ± SEM are shown; Statistical significance:
paired two-tailed t test. **F** IL-2Rα cell surface expression in OT-I WT (blue) and OT-I
*Srf*⁻/⁻(post) (red) cells 6 days after memory immune response, performed as in
(**D**).Top, representative plot of IL-2Rα expression. Bottom, data from 4 individual
mice after infection at day 6. Datapoints with Mean values ± SEM are shown. Statistical significance: paired t test. Source data are provided as a Source Data file.

ability to switch to high-affinity conformation upon TCR stimulation
remained intact (Fig. S7C, D). Moreover, cluster formation by OT-I
*Mrtfab*⁻/⁻ cells was not restored by exogenous IL-2, in contrast to STAT5
phosphorylation (Fig. 7C). Taken together these data are consistent
with a model in which MRTF-SRF-dependent clustering facilitates
receipt of paracrine IL-2 signals.

### Homotypic clustering of CD8⁺ T cells is MRTF-dependent in vivo
We next evaluated the role of MRTFs in CD8⁺ T-cell dynamics in vivo.
To synchronise T-cell activation as far as possible, we co-transferred
OT-I *Mrtfab*⁻/⁻ and OT-I WT CD8⁺ T cells into wildtype animals and
immunised them by injection with OVA peptides. IL-2Rα expression
was comparable in cells purified from lymph nodes 24 h following
activation (Fig. S7E). Multiphoton live imaging of popliteal lymph node
explants co-transferred CFSE- and SNARF-labelled cells showed that
although *Mrtfab*⁻/⁻ CD8⁺ T cells exhibited similar net displacement over
time, their speed was significantly reduced (Fig. S7F). To visualise cell
clustering in vivo, we generated LifeAct-GFP-expressing OT-I WT and
OT-I *Mrtfab*⁻/⁻ cells, labelled the OT-I WT cells with CellTrace Violet
prior to co-transfer, and analysed draining inguinal lymph nodes. Cells
of both genotypes were present in comparable numbers, and following
activation by anti-DNGR1-OVA and anti-CD40, many accumulated in
clusters of two cells or more (Fig. 7D). However, the proportion of OT-I
*Mrtfab*⁻/⁻ cells in clusters was significantly reduced (16.9 ± 2.1% versus
31.6 ± 2.8%; Fig. 7E), as was the frequency of homotypic interactions
between *Mrtfab*⁻/⁻ cells (17 ± 4.7% vs. 48.54 ± 6.8%) (Fig. 7E).

### *Mrtfab*-null CD8⁺ T cells exhibit defective cytoskeletal gene expression
MRTF-SRF signalling plays a central role in cytoskeletal gene
expression[34,41], so we next compared gene expression in wildtype and
*Mrtfab*-null OT-I CD8⁺ T cells. Since activated wildtype cells, but not
*Mrtfab*-null cells, can cluster and respond to endogenous IL-2, we
compared gene expression profiles in resting and activated cells. To
evaluate the effects of TCR activation and IL-2 stimulation separately,
naïve CD8⁺ T cells were activated by TCR crosslinking for 24 h, then
cultured for 16 h without TCR crosslinking in the presence of IL-12 to
maintain IL-2Rα expression ("TCR-activated/rested" cells), before stimulation with IL-2[53] (Fig. 8A).

RNAseq analysis identified 3039 genes that exhibited significant
differential expression in TCR-activated/rested *Mrtfab*-null cells compared to wildtype cells, regardless of IL-2 stimulation (Fig. 8B, Supplementary Data 1). These genes were grouped into 9 clusters (A-I) on
the basis of their normalised z-scored read counts (see Methods). Four
clusters (B,A,C, and I) encompassed 1134 genes whose transcript levels
were significantly reduced in *Mrtfab*-null cells (Fig. 8B). The 200 genes
in cluster B were not significantly induced by IL-2, but exhibited significantly impaired basal expression in TCR-activated/rested *Mrtfab*-
null cells. Cluster B also strongly overlaps with gene groups whose
induction upon TCR activation (Fig. S8A–C, cluster 10) or upon

activation and resting (Fig. S8A, D, E; clusters 9′,10′), was MRTF-
dependent.

Cluster B genes are enriched in Gene Ontology (GO) categories
related to actin cytoskeletal regulation, and reactome signalling
pathway components involved in rho-family GTPase signalling, infection, innate immunity, and immune cell biology (Fig. 8C, D; Supplementary Data 2). Cluster B includes genes demonstrated as MRTF-SRF
targets in other contexts, notably β- and γ-actin[41,42]. Both actins were
strongly induced transcriptionally in the transfer assay, which led to a
substantial increase in total actin protein expression; both basal and
induced actin protein levels were substantially reduced in *Mrtfab*⁻/⁻
cells (Fig. 8E, F). In contrast to Cluster B genes, clusters A,C, and I genes
are IL-2 inducible and were significantly enriched in gene ontology and
reactome categories related to immune cell and rho GTPase signalling
(Fig. 8C, D; Supplementary Data 2; see Discussion). A further 1369
genes whose expression increased in *Mrtfab*-null cells were significantly enriched in GO categories involved in various metabolic
processes (Fig. 8B–D, clusters E-H; Supplementary Data 2).

### MRTF-dependent F-actin assembly is required for IL-2 delivery
We next studied the role of cytoskeletal dynamics in cluster formation.
In wildtype cells, cluster formation by activated cells upon transfer to
culture without anti-CD3/CD28 TCR activation was abolished by
Latrunculin B (LatB), which also reduced STAT5 Y694 phosphorylation, indicating that F-actin assembly is required for both processes
(Fig. 9A, B). Activation of wildtype cells induced a substantial increase
in F-actin content, which was maintained during cluster formation, but
in *Mrtfab*⁻/⁻ cells, both basal and induced levels of F-actin were strongly
reduced (Figs. 9C and S9A).

To examine cluster formation in the absence of any confounding
effects arising from TCR engagement, we activated cells intracellularly
using PDBu and ionomycin. In this setting, cluster formation by wild-
type cells was dependent on LFA-1 and F-actin assembly, but IL-2
blockade only weakly inhibited cluster formation and STAT5 pY694
phosphorylation (Fig. S9B–E). However, although OT-I *Mrtfab*⁻/⁻
cells upregulated IL-2Rα, LFA-1, and ICAM-1 normally (Fig. S9B, C), cluster
formation was impaired: clusters contained fewer, less polarised cells,
were less densely packed, and contained almost 8-fold less F-actin
(Fig. 9D, E).

These data suggest that MRTF-SRF activity is required for cluster
formation per se rather than IL-2Rα expression or signal transduction.
We, therefore, tested whether MRTF activity affected the presentation
of IL-2 to cells within the cluster. To do this, we used the IL-2 "catch"
approach to trap IL-2 in close proximity to the cell surface[25]. Wildtype
or *Mrtfab*-null OT-I CD8⁺ T cells were coated with IL-2 catch reagent,
activated, and surface-proximal IL-2 detected 18 h later using fluorescently labeled IL-2 antibody. Activated OT-I WT cells exhibited large
puncta of IL-2 staining, reflecting retention of IL-2 within the cluster; in
contrast, the puncta observed in OT-I *Mrtfab*⁻/⁻ clusters were much
smaller (Fig. 9F). These results suggest that MRTF-dependent cluster

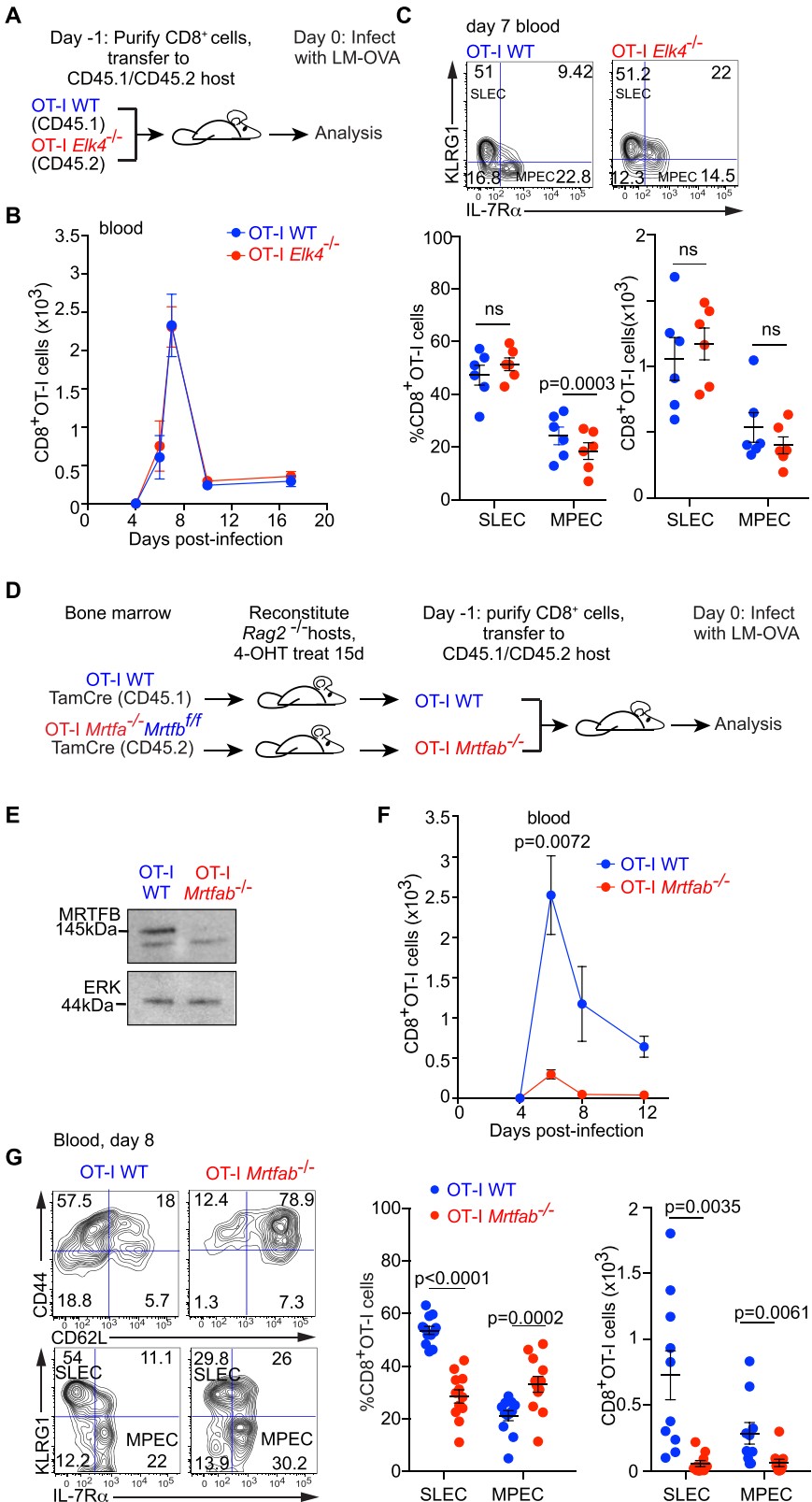

formation potentiates IL-2 signalling by facilitating IL-2 retention and/or presentation to cells in the cluster.

Previous studies have shown that at least some MRTF-dependent motility phenotypes reflect deficits in actin expression[43]. Given that actin expression is severely diminished in *Mrtfab*$^{-/-}$ cells we tested whether overexpression of actin could potentiate cluster formation. OT-I WT and OT-I *Mrtfab*$^{-/-}$ CD8$^+$ T cells were simultaneously activated

with PDBu/ionomycin and infected with lentivirus expressing mCherry or mCherry-β-actin. Expression of mCherry-β-actin both partially restored cluster formation and potentiated STAT5 Y694 phosphorylation in *Mrtfab*$^{-/-}$ cells, cluster numbers being similar to those of wildtype cells overexpressing mCherry-β-actin (Fig. 9G, H). Taken together these results support a model in which that the MRTF-SRF pathway potentiates paracrine presentation of IL-2 by facilitating

**Fig. 3 | The MRTFs mediate signalling to SRF in the CD8+ T-cell immune response. A** Experimental protocol for analysis of OT-I *Elk4*[-/-] CD8+ T cells. **B** Analysis by flow cytometry of expansion of 5000 adoptively co-transferred OT-I *Elk4*[-/-] and OT-I WT CD8+ cells after infection, mean value ± SEM (*n* = 6 mice). **C** Top, representative profile of cell surface expression of KLRG1 and IL-7Rα, 7 days post-infection; bottom, quantification of SLEC and MPEC. Data are mean values ± SEM for *n* = 6 mice represented by datapoints infected on the same day. Paired two-tailed t test is used for *p* value determination. **D** Irradiated *Rag2*[-/-] mice were reconstituted with bone marrow from OT-I WT TamCre or OT-I *Mrtfa*[-/-]*Mrtfb*[f/f] TamCre mice. Following reconstitution, chimeras were treated for 15 days with tamoxifen and OT-I WT (CD45.1) and OT-I *Mrtfab*[-/-] (CD45.2) naive CD8+ T cells were purified and co-transferred 1:1 into CD45.1/CD45.2 recipients, followed by rLM-OVA

infection the next day. **E** Immunoblot analysis before transfer reveals complete *Mrtfb* inactivation. Experiment has been done multiple times with similar results. **F** Analysis by flow cytometry of expansion of 5000 adoptively co-transferred OT-I WT and OT-I *Mrtfab*[-/-] CD8+ T cells following infection. Data show mean numbers ± SEM in the blood (20 μl). Data are representative of 2 independent experiments with 5 mice each. Statistical significance, paired two-tailed t test. **G** Cell surface expression of CD44, CD62L, KLRG1 and IL-7Rα by flow cytometry as in Fig. 1D. Quantification of SLEC and MPEC. Data show mean values ± SEM; datapoints represent individual mice pooled from 2 independent experiments with 6 mice each. Statistical significance, paired two-tailed t test. Source data are provided as a Source Data file.

---

homotypic cluster formation in activated CD8+ T cells, for which actin expression is in part limiting (see Discussion).

## Discussion

Here we show that the SRF transcription factor, acting cell-autonomously in partnership with its actin-regulated co-factors, the MRTFs, is essential for the effective response of CD8+ T cells to *L. monocytogenes* infection in the mouse. CD8+ T cells lacking the MRTFs or SRF undergo initial activation and proliferation, but subsequently fail to expand or to generate memory cells as a result of defective IL-2 signalling. Upon activation, *Mrtfab*-null cells induce IL-2Rα expression and secrete IL-2 normally, but generate aberrant homotypic clusters with reduced IL-2 retention. *Mrtfab*-null cells also exhibit substantial deficits in cytoskeletal gene expression. Cluster formation is dependent on F-actin assembly, and actin re-expression experiments demonstrate that aberrant cluster formation at least in part reflects limiting expression of cytoplasmic actin in *Mrtfab*-null cells.

We think it unlikely that TCF-SRF signalling contributes to the proliferative response to CD8+ T cells to LM-OVA infection, since *Mrftab*[-/-] cells, in which the TCF-SRF arm of the network remains intact, exhibit a proliferative defect similar to that of *Srf*[-/-] cells. It remains possible, however, that TCF-SRF signalling contributes to the infectious response in other immune cells. We also cannot rule out the possibility that TCF-SRF signalling contributes to other aspects of the response to LM-OVA infection. SAP-1/Elk4 functions redundantly with *Elk-1* in thymocyte development[37], and examination of multiply TCF-deficient animals may reveal such defects. The TCFs generally suppress MRTF-dependent SRF activity[42], but we did not observe increased expression of TCF-SRF controlled genes such as c-fos and egr-1, in *Mrtfab*[-/-] cells. This may reflect the relative expression of the TCFs and MRTFs, or the constitutively nuclear location of the TCFs. In humans, MKL1/MRTF-A inactivation causes susceptibility to bacterial infections[54,55]; the successful resolution of a chickenpox infection by one such patient might reflect residual MKL2/MRTF-B activity.

In the LM-OVA infection model, initial TCR activation and proliferation of *Srf*- and *Mrtf*-null CD8+ T cells was normal, but expansion was not sustained, and this reflected decreased proliferation rather than increased cell death. Differentiation into effector SLEC and MPEC remained intact, although there was a profound decrease in SLEC expansion resulting in skewing of CD8+ effector subsets towards MPEC formation. Related phenotypes have been observed in the LCMV and LM-OVA infection models upon inactivation of IL-2Rα[17–19], IRF4[22–24,56], LFA-1[26,28,29], ICAM-1[26,57] and PYK2[28]. *Srf*-null CD8+ T cells exhibited diminished granzyme B expression, although cytokine expression was not reduced, so it is possible that their cytotoxic function in vivo may be compromised. We have not examined this directly, but at least following activation in vitro, their ability to induce target cell killing was unaffected. We note that IRF4- and IL-2Rα-deficient CD8+ cells are defective in killing in vivo[19,23], although there are conflicting reports concerning LFA-1[29,58]. *Srf* is also required for the generation and/or persistence of memory cells, but *Srf*-null CD8+ T cells could be

generated by inactivation of *Srf* after priming. These cells, which expressed elevated levels of the Tcm markers CD62L, CCR7 and IL-2Rβ also exhibited defective sustained proliferation, indicating that SRF is also required for an effective CD8+ T memory cell recall response.

Previous experiments have shown that while the initial pro-liferative response of CD8+ T cells is cytokine-independent[11], sustained proliferation and SLEC accumulation require IL-2 signalling[17–19]. Even before the decline in proliferation seen in *Srf*- and *Mrtf*-null CD8+ T cells, signalling deficits are apparent, including reductions in IL-2Rα and IRF4 expression, reduced activity of the Akt-mTORC1-S6 kinase pathway, and reduced activation of the cytokine effector STAT5. IL-2, like SRF, is required for memory CD8+ T-cell generation and recall response[14,59]. Unlike *Il2*-null cells, however, *Srf*-null memory CD8+ T cells do not persist following primary infection. Given that *Srf*-null CD8+ T cells do generate MPECs, we speculate that SRF may have additional functions required for maintenance of the memory cell population, perhaps involving cell interactions (see below). Like both SRF and IL-2, the transcription factor IRF4 is dispensable for initial CD8+ T-cell activation and proliferation during infection, but necessary for sustained expansion and SLEC proliferation[22–24]. While initial IRF4 induction is TCR-mediated, *Irf4* is also IL-2 responsive[48] and it may be that IRF4 production during expansion is also IL-2 dependent.

What is the basis for the defect in IL-2 signalling? *Srf*- and *Mrtfab*-null CD8+ T cells produce and secrete normal amounts of IL-2 and other cytokines following activation, so it cannot reflect a secretion defect. Moreover, *Srf*- and *Mrtfab*-null cells maintain regulation of IL-2Rα expression, and retain the ability to respond to excess exogenous IL-2, indicating that the signal transduction apparatus is intact. During infection the response to IL-2, including STAT5 phosphorylation and proliferation, is facilitated by cell clustering and LFA-1/ICAM-1 interaction[25,60]. CD8+ T-cell clustering is LFA-1/ICAM-1 dependent[57,58] and LFA-1 and its effector kinase PYK2 play an important role in CD8+ T-cell proliferation and differentiation following LM-OVA or LCMV infection[26,28,29]. We found that following TCR activation in vitro, *Mrtfab*-null CD8+ T cells exhibited impaired homotypic clustering, forming smaller and less compact clusters with reduced F-actin content. *Mrtfab*-null CD8+ T-cell clusters also retain less IL-2. Activated *Mrtfab*-null CD8+ T-cell dynamics were also impaired in draining lymph node explant cultures: cells moved more slowly, and exhibited a decreased frequency of homotypic interactions. We propose that the LFA-1/ICAM-1-dependent CD8+ T-cell homotypic interactions required for effective IL-2 presentation in vivo in turn depend on the regulation of cytoskeletal dynamics through the MRTF-SRF axis. According to this view, defective IL-2 signalling reflects inefficient IL-2 presentation arising from defective cluster formation. The motility defects in *Mrtfab*-null may cells may also contribute to this by decreasing cell encounter frequency, but more work will be required to establish this.

Previous work has shown that paracrine sources of IL-2 are sufficient for effective proliferation in the LCMV model[59]. The cell-autonomous nature of the *Mrtfab*- and *Srf*-null phenotypes suggests that cytoskeletal defects in the recipient cell must be sufficient to impair IL-2 delivery through homotypic CD8+ T-cell interactions. It

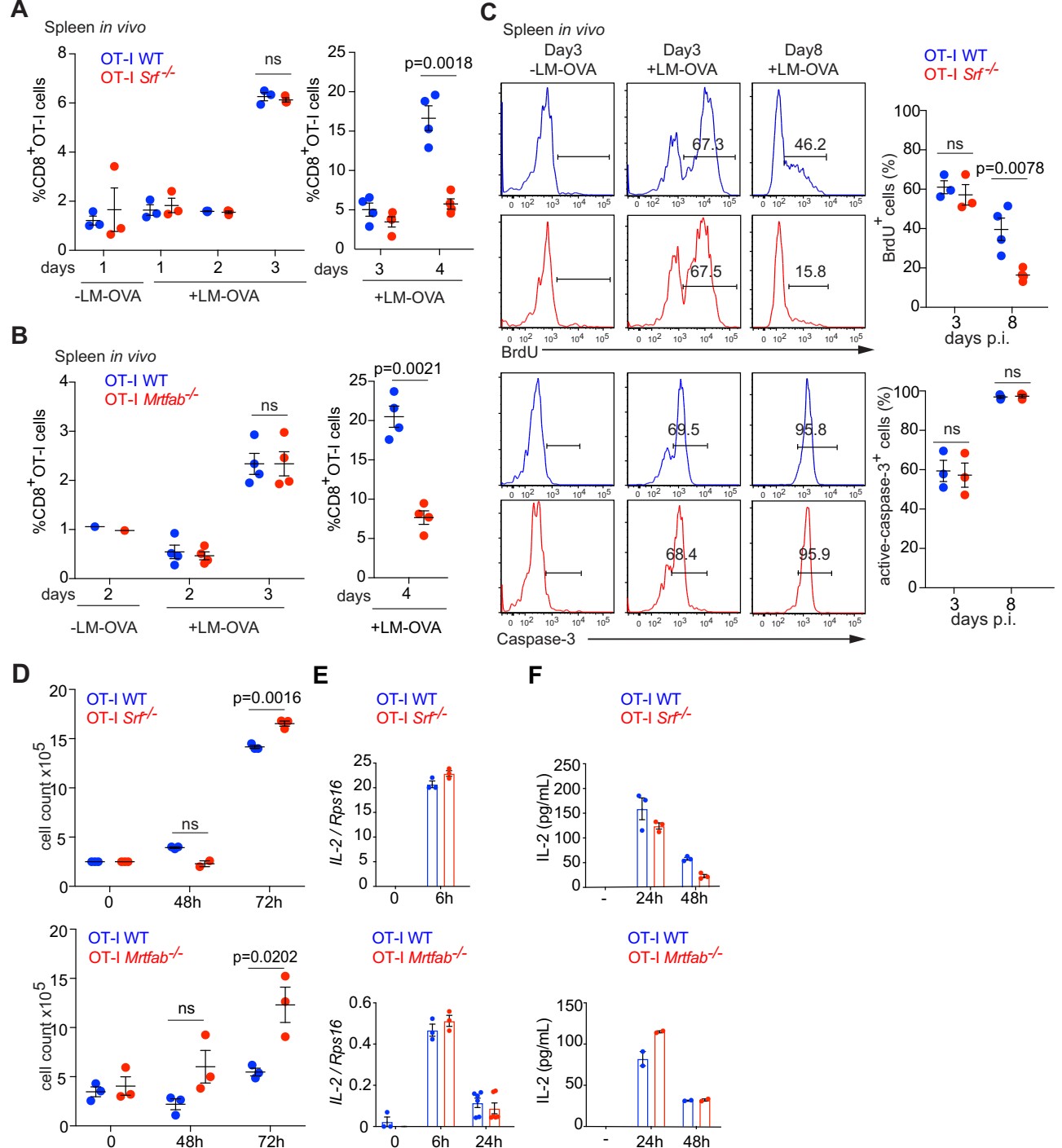

**Fig. 4 | Srf is required for the sustained proliferation of CD8+ T cells.**
**A** Frequencies of adoptively co-transferred (1 × 10⁶ cells) OT-I WT TamCre and OT-I *Srf*⁻/⁻ TamCre in spleens from uninfected animals compared with day 1 to day 4 post LM-OVA infection (left), or at days 3 and 4 post LM-OVA infection (right). Data show mean values ± SEM; datapoints represent individual mice, *n* = 3 mice (left) *n* = 4 mice (right). Representative of ≥ 2 experiments. Paired two-tailed t test is used for *p* value determination. **B** Frequencies of adoptively co-transferred (1 × 10⁶ cells) OT-I WT TamCre and OT-I *Mrtfab*⁻/⁻ TamCre in spleens from uninfected animals compared with day 2 to day 3 post LM-OVA infection (left), or at day 4 post LM-OVA infection (right). Data show mean values ± SEM; datapoints represent individual mice, *n* = 4 mice. Representative of ≥ 2 experiments. Paired two-tailed t test. **C** Upper panels, representative flow cytometry profiles of BrdU incorporation in OT-I WT and OT-I *Srf*⁻/⁻ in spleen labelled from days 1–3 and days 6-8 post-infection.

Lower panels, representative flow cytometry profiles of caspase-3 activity in OT-I WT and OT-I *Srf*⁻/⁻ cells at day 3 and day 8 post-infection, Quantitation is as (A). **D** Quantification of CFSE-labelling profiles of MACS-purified CD8 cells, either resting or activated with plate-bound anti-CD3/CD28 (5 µg/ml) for the indicated times. A representative of three independent experiments is shown; datapoints show mean of triplicate determinations ± SEM. For CFSE profiles see Fig. S4C. Unpaired two-tailed t test is used for *p* value determination. **E** Plate-bound anti-CD3/CD28 TCR-induced transcription of IL-2 mRNA in sorted OT-I lymph node cells. A representative of three independent experiments is shown. Datapoints show mean of triplicate determinations ± SEM. **F** IL-2 production measured by ELISA in sorted CD8⁺ T lymph node cells following activation by plate-bound anti-CD3/anti-CD28. Data are triplicate assays (±SEM) from one of two independent experiments. Source data are provided as a Source Data file.

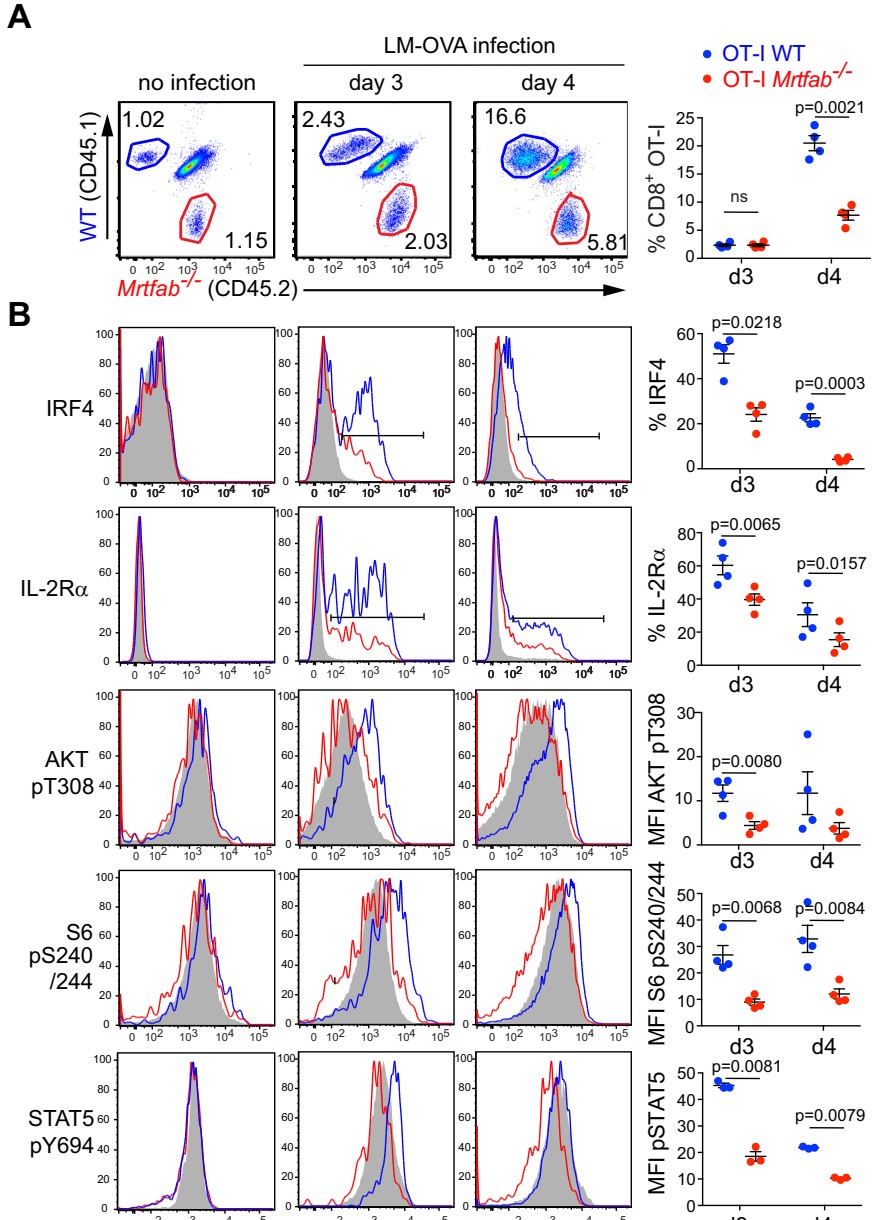

**Fig. 5 | Expansion defects in Mrtfab- and Srf-null CD8+ cells are accompanied by defects in IL-2 signalling. A** Flow cytometry analysis of splenocytes from mice co-transferred with OT-I WT CD45.1 and OT-I *Mrtfab*−/− CD45.2 (1.10⁶ cells) and harvested at day 3 or 4 post-infection. Cells were stained with antibodies to CD8, CD45.1 and CD45.2 and CD8⁺ gated adoptive populations CD45.1 OT-I WT (blue) and CD45.2 OT-I *Mrtfab*−/− (red) were analysed. A representative of 3 independent experiments is shown. Each data point represents a single mouse (*n* = 4). Mean values ± SEM, with statistical significance by paired two-tailed t test. **B** Relative

levels of extracellular CD25 (IL-2Rα) protein expression, intracellular IRF4, pAKTthr308, pS6 240/244, and pSTAT5 proteins from the populations in (**A**): OT-I WT (blue), OT-I *Mrtfab*−/− (red) and endogenous CD45.1/CD45.2 CD8⁺ T cells (grey). Graphs show quantitations of protein expression by frequencies or MFI. Each data point represents a single mouse (*n* = 4 mice) in a representative of at least 3 independent experiments per condition. Data show mean values ± SEM, with statistical significance by paired two-tailed t test. Source data are provided as a Source Data file.

therefore remains possible that *Mrtfab* and *Srf* inactivation will also impair delivery of other cytokines, such as IL-12 and interferons, that is mediated by heterotypic interactions[26,51].

The immune synapse is also LFA-1/ICAM-1-dependent. However, *Srf* or *Mrtfab* inactivation did not detectably impair initial CD8⁺ T-cell activation in the LM-OVA model, indicating that naive cells are capable of forming a functional synapse. This may reflect other adhesion mechanisms operating in this context. Indeed, following TCR ligation in vitro, activated *Srf*- or *Mrtfab*-null CD8⁺ T cells actually exhibited enhanced proliferation, and we speculate that cytoskeletal disruption resulting from MRTF-SRF inactivation might alter the dynamics of TCR

signalling complexes in this setting. In the LM-OVA model, however, initial proliferation of *Srf*- or *Mrtfab*-null cells remained normal. Thymocyte positive selection, which also requires TCR signalling, is also unaffected in MRTF-deficient animals (PC, DM and RT, in preparation).

In addition to providing a homeostatic feedback loop that controls actin gene expression[34,39,61], the MRTFs also control numerous genes controlling F-actin assembly and stability[33,34,41]. MRTF-SRF signalling is thus required for motility, adhesion, and other cytoskeletal functions[32,40,47] (Fig S9G). Nevertheless, at least in some contexts actin re-expression is sufficient to suppress the motility defects associated with MRTF inactivation[43]. In CD8⁺ T cells, MRTF inactivation reduced

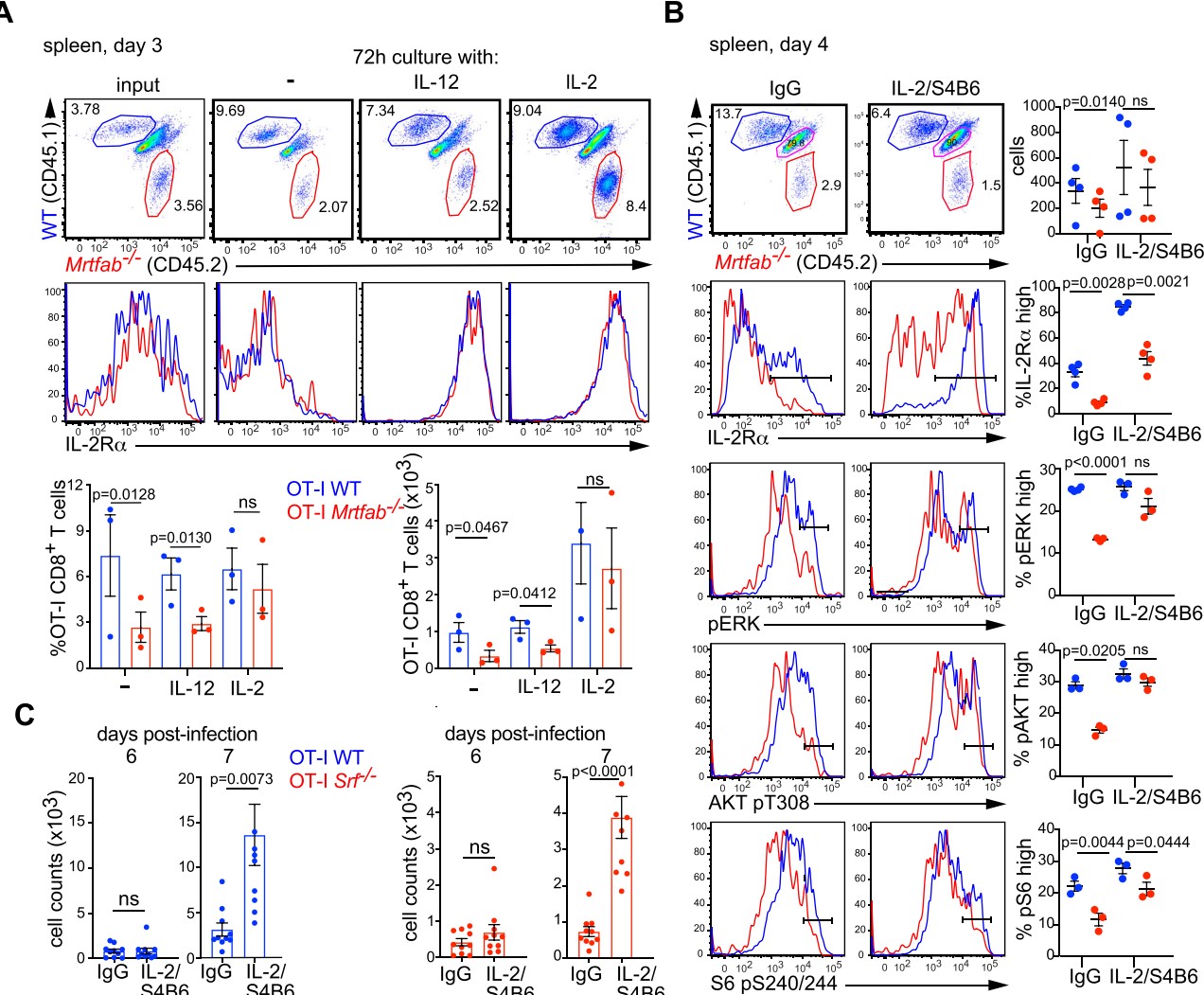

**Fig. 6 | Srf- and Mrtfab-null cells remain responsive to exogenous IL-2. A** Co-transferred OT-I WT and OT-I *Mrtfab⁻/⁻* cells were harvested from spleen 3 days post LM-OVA infection (input) and cultured a further 3 days without cytokine, with IL-12 or IL-2. Top panels, dot-plots of CD8⁺ gated splenocytes, with histograms indicating the amount of IL-2Rα (CD25) expression. Bottom, relative proportions and numbers of OT-I WT and OT-I *Mrtfab⁻/⁻* cells after 72 h culture (mean ± SEM, *n* = 3 mice), paired two-tailed t test. This is a representative experiment out of 2 independent experiments with ≥3 mice each. **B** Potentiation of OT-I *Mrtfab⁻/⁻* signalling in vivo by administration of IL-2/S4B6 complexes. Mice were co-injected with 1 × 10⁶ each OT-I WT and OT-I *Mrtfab⁻/⁻* CD8⁺ T cells and infected with Listeria the following day. Mice were treated with either IL-2/S4B6 complexes or control IgG from day 1 to 3 post-infection. Dot-plots of CD8-gated splenocytes at day 4 post-infection show relative proportions of OT-I wildtype (CD45.1) and OT-I *Mrtfab⁻/⁻* (CD45.2) cells. Histograms show IL-2Rα expression, pERK, AKT pT308 and S6 pS240/244. Right: quantification with each dot representing a mouse, *n* = 3–4 mice, paired two-tailed t test. Data are shown as mean ± SEM. **C** Potentiation of OT-I WT and OT-I *Srf⁻/⁻* proliferation in vivo by administration of IL-2/S4B6 complexes from day 3 to 5 post-infection. Mice were injected with 5000 cells each OT-I WT and OT-I *Srf⁻/⁻* CD8⁺ T cells, infected with Listeria the following day, and cell numbers in the blood are quantified at 6 and 7 days after infection. Data are mean numbers ± SEM in blood (20 µl), 10 mice per time point, unpaired two-tailed t test. Source data are provided as a Source Data file.

β- and γ-actin transcription, causing a pronounced deficit in both actin expression and F-actin assembly, which is required for homotypic cluster formation. Strikingly, β-actin re-expression at least partially restored cluster formation in this system (the two vertebrate cytoplasmic actins are largely functionally equivalent[62]). Together with previous findings, this suggests that the co-option of actin regulators as targets of MRTF-SRF signalling may represent an evolutionary elaboration of the basic actin homeostatic feedback loop, analogous to the acquisition of "feed-forward" pathways in gene expression circuits[63].

In conclusion, we have shown that SRF, acting with its MRTF co-factors, is essential for the effective immune response of CD8⁺ T cells during LM-OVA infection in the mouse. Our observations demonstrate the critical importance of homotypic clustering for cytokine signalling, and reveal the role played by MRTF-dependent cytoskeletal

dynamics in the control of cell proliferation by IL-2 during the immune response.

## Methods

### Ethics statement

Animal experimentation was carried out in accordance with the UK Animals (Scientific Procedures) Act 1986, under UK Home Office project licences PPL PP0389970, P7C307997, 80/2602 and 70/7982. All project licence applications were approved by the Francis Crick Institute Project licence review subcommittee of the Crick's Animal Welfare and Ethical Review Body. The UK Home Office accredited all researchers for animal handling and experimentation. Dispensation to carry out animal research at the Francis Crick Institute was approved by the Institutional Ethical Review Body and granted by the UK government Home Office.

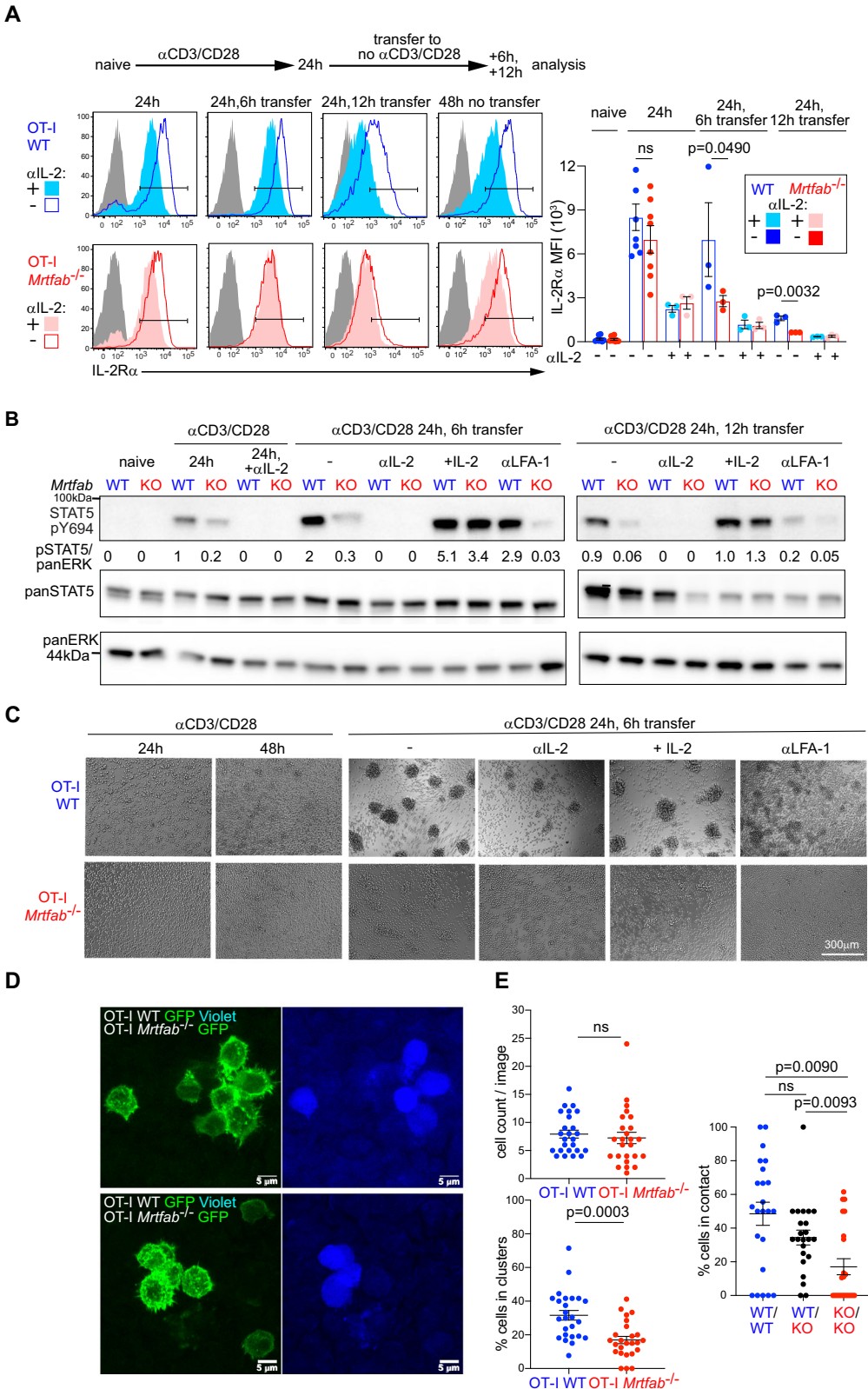

## Mice

C57BL6/J mice were *Elk4*[L−/−35,37] (SAP-1 null); R26CreERT2 (*tm9(cre/ESR1)Arte*[64]); conditional *Srf*[f/f65]; conditional *Mrtfa−/−Mrtfb*[f/f 47] and transgenic Lifeact-EGFP[66]. *Srf*[f/f] TamCre (CD45.2), WT TamCre (CD45.1), *Mrtfa−/−Mrtfb*[f/f] TamCre (CD45.2), *Elk4−/−* and Lifeact-EGFP animals were crossed to OT-I TCR transgenics (*Tg(TcraTcrb)1100Mjb*/Crl). OT-I Lifeact-EGFP animals were further crossed to *Mrtfa−/−Mrtfb*[f/f] TamCre. All

strains were bred separately. All genetically modified mouse lines were back-crossed to C57BL/6 J. For all experiments young adult female mice were used, aged between 8-12 weeks old. For reconstitution, one-week acid-watered *Rag2−/−* (RAG-2 tm1Fwa) hosts were [137]Cs-irradiated (2 × 5 Gy) and 24 hours later, bone marrow cells injected into the tail vein. To induce CreER[T2], mice were fed powdered tamoxifen pellets (Envigo) mixed 4:1 with normal diet to ease

**Fig. 7 | Deficient IL-2 signalling in activated Mrtfab⁻/⁻ OT-I cells correlates with defective homotypic clustering. A** MACS-purified OT-I WT and OT-I *Mrtfab*⁻/⁻ CD8⁺ T cells from tamoxifen-fed reconstituted mice activated in vitro with plate-bound anti-CD3/anti-CD28 (5 μg/ml) with or without anti-IL-2 blocking antibody (JES6-1A12)(10 μg/ml) for the times indicated. 24 h post TCR/CD28 activation, cells were transferred or not with supernatant to uncoated wells for an additional 6 or 12 hours. Cells were stained for IL-2Rα and CD8 and fixed. Left: Histograms of OT-I WT (blue) and OT-I *Mrtfab*⁻/⁻ (red).Right: quantification of IL-2Rα MFI ± SEM for at least 3 independent experiments or purification represented as data point. Unpaired two-tailed t test is used. **B** Stat5 phosphorylation (pSTAT5 Y694) ± anti-IL-2 blocking antibody (JES6-1A12) (10 μg/ml), recombinant mIL2(20 ng/ml) or anti-CD11a/ anti-LFA-1 blocking antibody (10 μg/ml). Quantification below (pStat5/panErk). Data presented is representative of at least 3 biological replicates across all conditions. **C** Brightfield images of purified OT-I WT and OT-I *Mrtfab*⁻/⁻ CD8+ generated and cultured as in (**A**) and (**B**). Images are representative of at least 3 independent experiments. Scale bar, 300 μm. **D** *Mrtfab*⁻/⁻ cells exhibit defective clustering in vivo. OT-I WT Lifeact-GFP (CellTrace violet-labelled) and OT-I *Mrtfab*⁻/⁻ Lifeact-GFP were co-injected at 1:1 ratio i.v. into WT mice, and immunised the next day with DNGR1-OVA and anti-CD40 by subcutaneous injection into both hocks. The draining inguinal lymph nodes were isolated 24 h later for confocal microscopy. Representative images with OT-I WT and OT-I *Mrtfab*⁻/⁻ Lifeact-GFP (green) (left) and OT-I-WT cellTrace violet (right). Regions containing clusters demonstrate the majority of the cells participating in clusters (≥ 2 cells) to be OT-I WT Life-actGFP. **E** Quantitation of the experiment presented in (**D**). Top left, numbers of GFP+ cells present in each region analysed (*n* = 25 images); bottom left, percentage of OT-I-WT or OT-I *Mrtfab*⁻/⁻ cells in clusters out of total cells per image (*n* = 25 images); dots represent data from each image. Right, mean frequencies of WT-WT, WT-KO and KO-KO contacts in two independent experiments. Data are mean values ± SEM; Statistical significance, paired two-tailed t test. Source data are provided as a Source Data file.

absorption for 15 days. Animals were maintained under specific-pathogen-free conditions in The Francis Crick Institute UK Biological Resources Facility. Housing conditions were: light cycles fluctuating from 7 am to 7 pm, temperature range 20–24 degrees C, 55% ± 10% humidity. Animals were euthanised by $CO_2$ asphyxiation and cervical dislocation.

### Infection
*Listeria monocytogenes* expressing ovalbumin (rLM-OVA)[45] was cultured overnight in brain-heart infusion medium (Fluka 53286-5006) with erythromycin, re-seeded and cultured to $OD_{600}$ of 0.1–0.2, and diluted in PBS to $10^4$ bacteria/ml for injection. Splenocytes or lymph node cells were enriched for CD8⁺ T cells by negative selection using a CD8 T-cell isolation kit (Miltenyi Biotech) based on magnetic cell separation (MACS). 5000 cells (low input) or $2 \times 10^6$ cells (high input) and adoptively transferred by injection into the tail vein. The next day, mice were infected intravenously with 2000 CFU. For recall response, memory CD8⁺ T cells purified by flowsort to 99% purity from splenocytes immunostained with CD8, CD45.1 and CD45.2 were adoptively co-transferred (1:1, $5 \times 10^5$ cells) into naive syngeneic mice followed by infection with LM-OVA and analysis 6 days later.

### Immunisation
For cell activation and ex vivo analysis of cell clustering, mice were immunised with VacOva (vac-pova, InvivoGen; 50 μg/mouse) and LPS (L6529 Sigma; 30 μg) co-injected subcutaneously into the hock; and for live microscopy, with anti-DNGR1 397-OVA fusion protein[67] (2 μg; gift from Sandra Diebold) and anti-CD40 (1C10; 25 μg) co-injected into the footpad.

### Flow cytometry
Tissues were disaggregated through a 70 μm nylon mesh in cold RPMI 10% FCS. Blood was collected into Sarstedt EDTA KE/1.3 tubes before red blood cell lysis. Cells were stained in ice-cold FACS buffer (1% FCS, 2 mM EDTA in PBS) with combinations of fluorochrome-conjugated antibodies and analysed using BD-Fortessa Instruments and FlowJo 9.9 software. For intracellular staining, cells were fixed and permeabilised using eBioscience intracellular fixation and permeabilisation kit (BD Biosciences). For pSTAT5 and pERK staining, cells were fixed in 2% paraformaldehyde, washed and permeabilised in ice-cold methanol for 30 min, washed twice in PBS, 10% FCS and stained for 1 h. Samples were run on LSR-IIB or Fortessa II (BD Biosciences) and analysed with FlowJo software (BD Biosciences). Gating strategies for flow cytometry are summarised in Fig. S10.

### Cell isolation
Where indicated, T cells were sorted to >97% purity as judged by cell surface marker expression on a BD INFLUX or FACS ARIA-III using BD FACS Diva software for subsequent analysis. Coulter CC Size standard beads (Beckman Coulter) were used for calculating cell numbers. For clustering experiments, CD8⁺ naive T cells were enriched from lymph node or spleen of bone marrow-reconstituted mice by MACS.

### Cell stimulation
Cells were cultured in R10 medium (RPMI 1640 containing 10% heat-inactivated FCS, 50 μM 2-mercaptoethanol with penicillin and streptomycin; Sigma-Aldrich) and incubated at 37 °C, 10% $CO_2$. For cell proliferation assays, purified naive CD8⁺ T cells were labelled with CFSE (5 μM, Molecular Probes) for 15 min in RPMI 1640 without serum and washed in R10 medium to stop the reaction. Cells were cultured on plate-bound anti-CD3 (2C11) and anti-CD28 (37.51) (5 μg/ml) or stimulated with OVA peptides of decreasing affinity (N4 > Q4 > V4) for the OT-I TCR: N4 (SIINFEKL), Q4 (SIIQFEKL) and V4 (SIIVFEKL)[68].

For intracellular cytokine staining, cells were stimulated with 10 nM SIINFEKL (OVA) peptide for 5 h at 37 °C in the presence of Brefeldin A (Sigma-Aldrich) for the last 2 h before staining. For cytokine release, secretion of IL-2 was measured by IL-2 ELISA (M2000 R&D); secretion of IFN-γ and TNF-α was measured by mouse Interferon gamma ELISA kit (AB100689, Abcam) and mouse TNF alpha ELISA kit (AB208348, Abcam).

For ex vivo in vitro culture, splenocytes were harvested from 3-day post-infection mice and cultured for a further 72 h at 1–2 × 10⁶ cells/ml in media alone, or supplemented with 20 ng/ml rmIL-2 or 20 ng/ml rmIL-12 before analysis by flow cytometry.

### In vivo IL-2 modulation
IL-2/mAb complexes were generated by incubating murine rIL-2 (rmIL-2, Immunotools) with S4B6 anti-IL-2 monoclonal antibody at a 2:1 molar ratio (1.5 μg/ml IL-2, 50 μg/ml S4B6) for 15 minutes at room temperature. IL-2/S4B6 complexes or IgG2α control antibodies (both eBioscience) were injected i.p.

### In vivo T-T clustering assay
Purified OT-I Lifeact-EGFP CD8⁺ T cells were labelled with 5 μM of CellTrace Violet for 30 min at 37 °C and then transferred into recipient mice by tail vein injection. Mice were immunised a day later and the draining lymph nodes were harvested 24 hours later. Lymph nodes were in fixed in 4% PFA overnight at 4 °C and then transferred to a solution of 30% sucrose in 1× PBS at 4 °C overnight. Lymph nodes were then embedded, snap frozen in OCT (VWR 361603E) and stored at −80 °C. 30 μm tissue sections were produced and mounted into Pro-Long Diamond Antifade (Thermofisher). Sections were scanned using a Zeiss LSM880 inverted with an airyscan module. (alpha Plan-apochromat ×63/1.46 Oil Korr M27 objective lens, 0.5 μM intervals, x, y: 0.033 μm). Distance between cells and number of direct connections between cells were measured using a custom Cellprofiler pipeline. Briefly, multichannel confocal images were split into single-channel images and processed for background removal in FIJI. Cells were

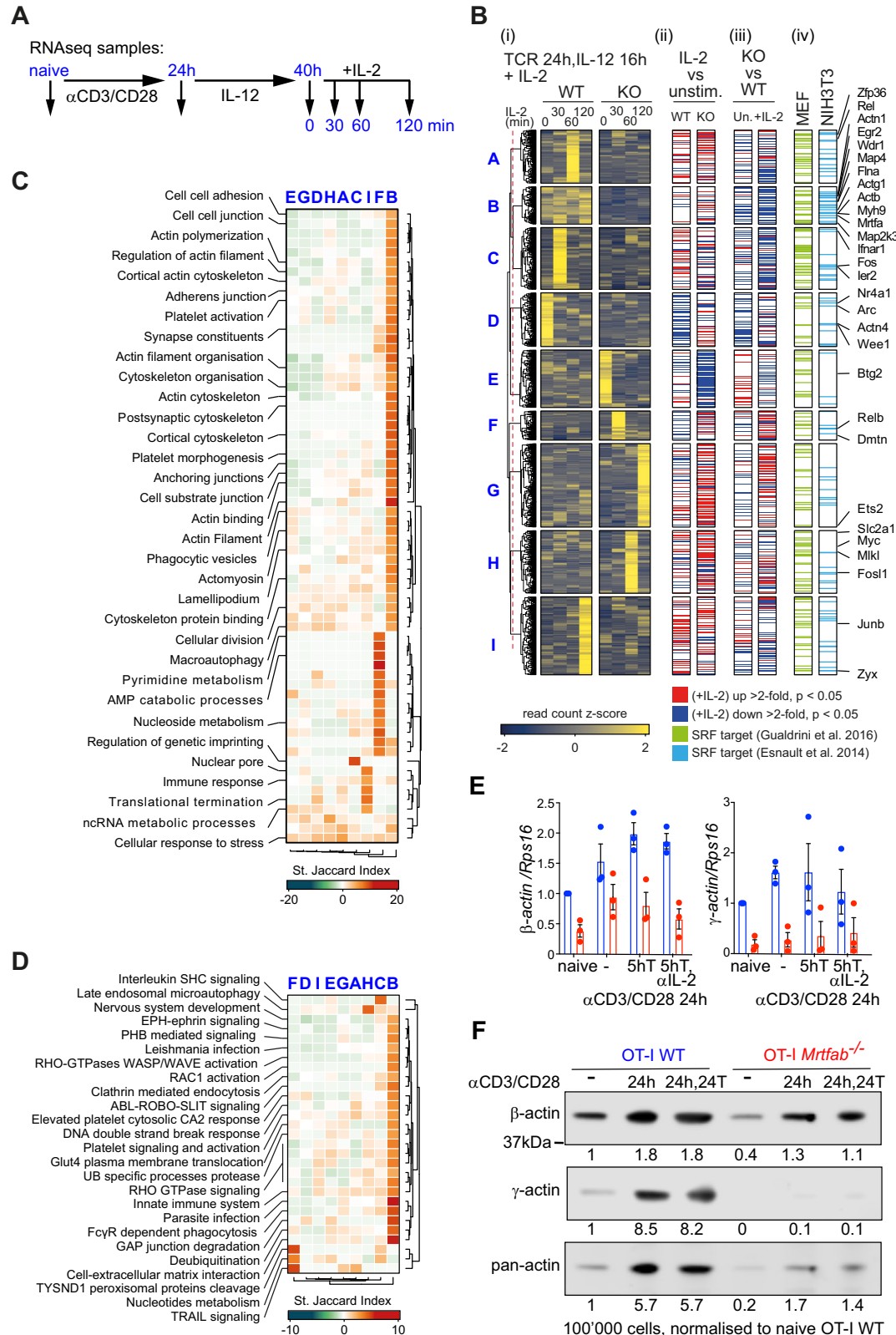

**A** RNAseq samples: naive → 24h αCD3/CD28 → 40h IL-12 → +IL-2 at 0, 30, 60, 120 min

**B** (i) TCR 24h, IL-12 16h + IL-2; (ii) IL-2 vs unstim.; (iii) KO vs WT; (iv)

read count z-score: −2 to 2

(+IL-2) up >2-fold, p < 0.05
(+IL-2) down >2-fold, p < 0.05
SRF target (Gualdrini et al. 2016)
SRF target (Esnault et al. 2014)

**C** St. Jaccard Index −20 to 20

**D** St. Jaccard Index −10 to 10

**E** β-actin/Rps16; γ-actin/Rps16

**F** OT-I WT / OT-I Mrtfab⁻/⁻; β-actin, γ-actin, pan-actin. 100'000 cells, normalised to naive OT-I WT

manually segmented using the FIJI Labkit plugin using the GFP channel. The output of the Labkit segmentation produced an image in which each segmented cell had a unique integer pixel value. Segmentation and individual channel images were then imported in CellProfiler v4.2.4, the segmented cells were converted into 'Objects' using the ConvertImageToObject module. The fluorescence intensity of the Violet channel was measured inside each object using the

MeasureObjectIntensity module and used to filter the objects into the Violet positive and Violet negative categories using the FilterObject module. The distance between cells and the number of neighbouring cells within a 10pixel radius was then measured using the MeasureObjectNeighbors module. Note that expression of LifeactGFP is lower in Mrtfab-null cells because its expression is controlled by CAG promoter, which is MRTF-dependent[69].

**Fig. 8 | Gene expression deficits in OT-I *Mrtfab⁻/⁻* cells.** See also Supplementary Data 1 and 2. **A** Purified naive wildtype or *Mrtfab*-null OT-I CD8⁺ T cells were stimulated as shown[53]. RNAseq was carried out on naive cells, cells activated by TCR crosslinking for 24 h, or activated cells cultured in IL-12 for 16 h ("activated/rested cells") and then stimulated with IL-2 for various times. Data show mean values ± SEM of three biological replicates, each representing cells purified from pooled lymph nodes from individual animals. **B** (i) Normalised z-scored read counts of activated/rested cells stimulated with IL-2, grouped according to gene expression patterns by unsupervised clustering. (ii) Genes showing significant changes upon IL-2 stimulation at any time point in wildtype (left) and *Mrtfab*-null cells (right). Genes showing an absolute fold-change greater than 2 at p < 0.05 (DESeq2) are color-coded. Fold-change and padj values are reported in Supplementary Data 1. (iii) Genes whose expression is impaired by MRTF inactivation, in activated/rested cells (left) or IL-2 stimulated cells (right), displayed as in (ii). (iv) Genes identified as candidate SRF targets in TPA-stimulated MEFs[42] or serum-stimulated NIH3T3 fibroblasts[41]. **C** Gene ontology categories significantly overrepresented (p < 0.01) in the clusters identified in (**B**) are summarised by standard Jaccard index scores. Significance was established by hypergeometric testing using the "phyper" function from the R library "stats" and *p* values were adjusted for multiple comparisons using

the "p.adjust" function with the Benjamini & Hochberg method from the same library. St JI scores and adjusted *p* values are reported in Supplementary Data 2. Gene ontology categories significantly overrepresented (P < 0.01) in the clusters identified in (**B**), summarised by Standard Jaccard Index. **D** Reactome pathway categories (https://reactome.org) significantly overrepresented (P < 0.01) in the clusters identified in (**B**) are summarised by Standard Jaccard Index scores, with significance established and reported as in (**C**). **E** Expression of β- and γ-actin in MACS-purified OT-I WT and OT-I *Mrtfab⁻/⁻* lymph node cells following TCR activation assessed by qRT-PCR. MACS-purified OT-I WT and OT-I *Mrtfab⁻/⁻* cells from the spleens of tamoxifen-fed bone marrow-reconstituted mice were activated in vitro with plate-bound anti-CD3/CD28 (5 μg/ml) for 24 h, then transferred with supernatant to uncoated wells and cultured 5 h with or without anti-IL-2 blocking antibody. Data show mean values ± SEM of three independent experiments, normalised to expression in naive OT-I WT cells. **F** Immunoblot analysis of β-actin, γ-actin, and total actin expression in MACS-purified OT-I WT and OT-I *Mrtfab⁻/⁻* lymph node cells following TCR activation. Cells were activated in vitro as in (**E**) and transferred with supernatant to uncoated wells for a further 24 h. Protein levels were normalised to those of naive cells. Source data are provided as a Source Data file.

## Immunohistochemistry

For staining of spleen sections, OCT embedded fixed spleens injected with either OT-I WT or OT-I *Mrtfab⁻/⁻* T cells labelled with cell tracker deep red dye were prepared as above. 8 μm frozen sections were air dried for 30 min at room temperature in the dark and rehydrated for 3 min in PBS. Sections were blocked and permeabilised in blocking buffer 0.3% Triton X-100, 3% BSA in PBS for 1 hour at room temperature followed by incubation with primary antibody diluted in blocking buffer overnight at 4 °C. After 3× washes in 1× PBS, sections were incubated for 1 hour with secondary antibody, followed by 3 washes in PBS and mounting with ProLong Diamond Antifade.

## Multiphoton live microscopy of explanted popliteal lymph nodes

For analysis of cell migration in vivo, *Mrtfab⁻/⁻* and WT OT-I cells were labelled with 2 μm CFSE (C34554 Invitrogen) or 6 μM SNARF (S22801 Invitrogen) and $2 \times 10^6$ cells mixed in 1:1 ratio were transferred into CD45.1/CD45.2 recipients. One day following immunisation, popliteal draining lymph nodes were removed and immobilised on coverslips with hilum facing away from the objective. Lymph nodes were continuously perfused with warmed (37 °C) RPMI 1640 medium without Phenol Red (GIBCO, Invitrogen) bubbled with carbogen (95% O₂ 5% CO₂). Images of live cells were taken using a LSM Zeiss 710 multiphoton, ×20 NA 1.05 water immersion objective and a pulsed Ti:sapphire laser (Spectra Physics MaiTai HP DeepSee) voxel size:$1.6605 \times 1.6605 \times 3$ μm³, to a depth of about 100 μm from the capsule. Tracking of live cells was analysed with Imaris 9.5 using the spot tool applying gaussian filter (2 μm) and background subtraction (6 μm). Cells that could be detected continuously for 15 min were analysed and the average speed and total displacement were measured.

## In vitro T-T clustering assays

Antibody-induced clusters were generated by culturing MACS-purified naive OT-I T cells on plate-bound anti-CD3 and anti-CD28 (5 μg/ml each) for 24 hrs in the presence or absence of anti-IL-2 blocking antibody (JES6-1A12) (10 μg/ml), before transfer of cells and supernatant to non-coated wells with or without anti-CD11a/LFA-1 (M17.4)(10 μg/ml) blocking antibody and recombinant mIL2 (20 ng/ml) or Latrunculin B (5 μM) for the times indicated. Pharmacologically-induced clusters were generated by culturing T cells in 25 ng/ml Phorbol 12,13-dibutyrate (PDBu) (P1269 Sigma) and 500 ng/ml ionomycin (I0634 Sigma) for 18 h. Clusters were then cultured for an additional 5 hours with or without anti-IL-2 or anti-LFA-1 blocking antibodies or with mIL2 (20 ng/ml) for the last 15 min or with Pyk2/FAK inhibitor PF431496 (5 μM) for

the last 1 h. Following incubation with anti-IL-2 blocking antibody some cells were restimulated with recombinant human IL-2 (Sigma) for 15 min. For G- and F-actin quantitation, cells were either analysed by flow cytometry after staining with Alexa Fluor 488 DnaseI (Invitrogen D12371) or Alexa Fluor 647 Phalloidin (Invitrogen A22287), or snap frozen and analysed using by sedimentation assay for G- and F-actin as described[70].

## Cluster imaging

Clusters were visualised using ZEISS Observer D1 AxioCam. For fluorescence staining, activated T cells were transferred to poly-L-lysine-coated Matek dishes (50 μg/ml), incubated 15 min at 37 ºC with 5% CO₂ and fixed with 2% PFA for 10 min. Clusters were washed in PBS, blocked in 3% BSA, permeabilised in 0.2%Triton and then immunostained with anti-pericentrin and counterstained with Texas Red phalloidin and DAPI. For the IL-2 catch assay, purified CD8⁺ T cells were coated with mouse IL-2 catch reagent from a mouse IL-2 secretion assay detection kit (Miltenyi Biotec) by incubating T cells in a 20× dilution of IL-2 catch reagent in R10 medium for 15 min on ice, washing once in R10 medium and then stimulating with PDBu/Iono to form T-cell clusters as above. After fixation, cells were stained with phycoerythrin-conjugated IL-2 detection antibody (1:20 dilution in R10 medium).

Stained clusters were mounted in Mowiol with coverslips and analysed by confocal microscopy (Zeiss LSM710 inverted). Single-plane brightfield images and Z stacks of the appropriate channel of fluorescent images (0.37 μm intervals) were acquired with a Plan-Apochromat 40x/1.3 oil Ph3 M27 Objective lens. Acquired images were analysed using Imaris 9.6. Images were processed using Imaris background subtraction filter for each channel. Additionally, a gaussian and median filter was applied on the channel of interest. For in vitro generated cluster, cells were detected using the Imaris spot tool on the DAPI channel, distance between cells was measured using the spots positions. The mean intensity of the phalloidin staining and the cell sphericity were measured for each individual cell using the Imaris cell detection tool. For the IL-2 catch analysis a median filter was first applied on the images.The Imaris surface tool was used with two volume thresholds (5.04 and 10.1 μm³) to create three colour-coded classes of volumes of IL-2 signals in each cluster.

## Lentivirus infection

β-actin was expressed as a fusion with mCherry using the pLVX-mCherry lentiviral vector (gift from Jasmine Abella); control virus expressed mCherry alone. Viral particles were obtained by transfection of Phoenix cells with DNA mix (0.6 μg VSVG (envelope), 3 μg px PAX2 (packaging), 6 μg pLVX derivative in 300 μL of transfection reagent

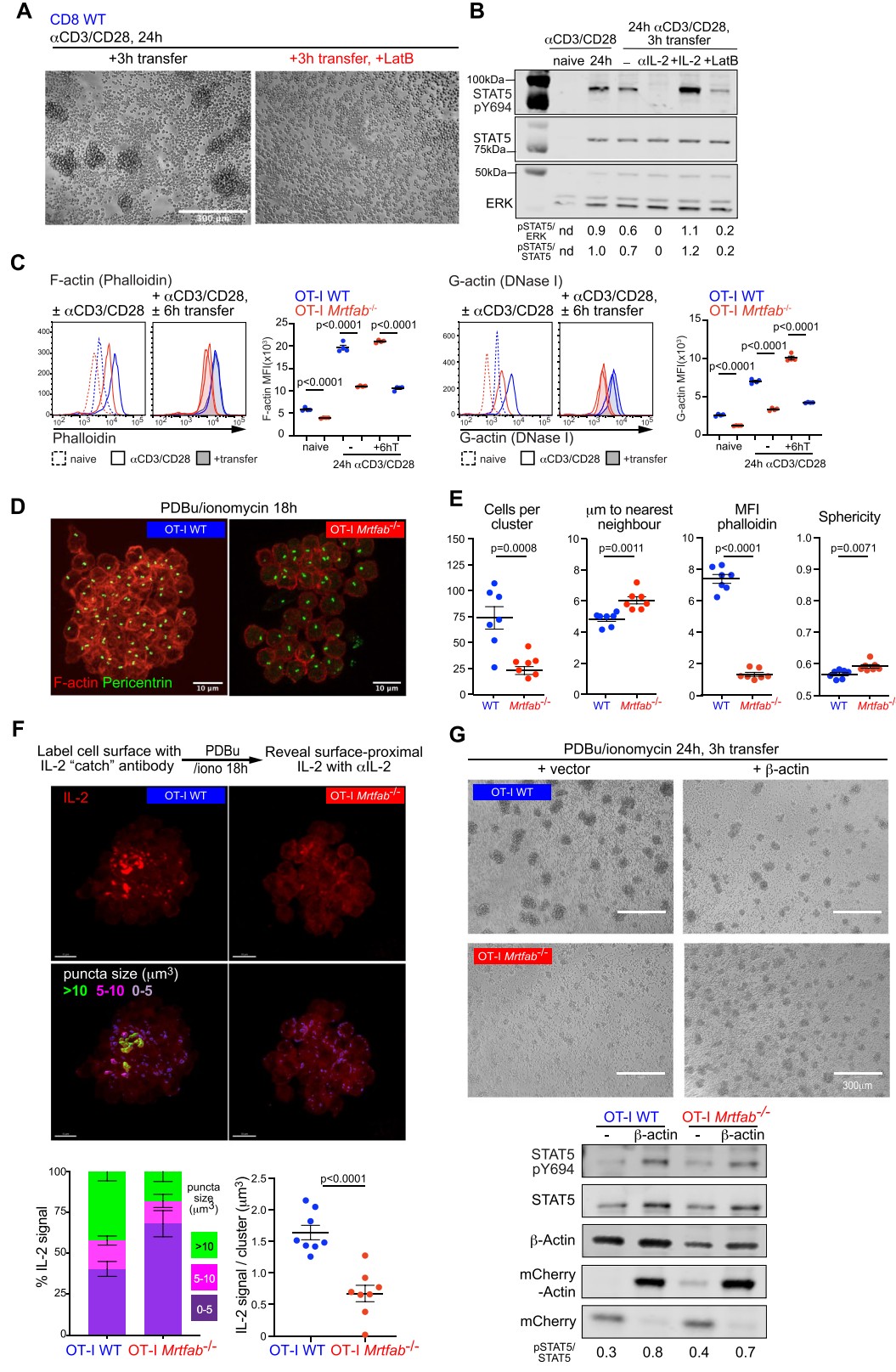

containing 42 μL Fugene with 258 μL opti-MEM/dish and with a Fugene:DNA(pTripz) ratio of 7:1. MACS-purified *Mrtfab⁻/⁻* or WT OT-I Tam Cre naive cells (1.10⁶ cells) were infected with control or β-actin virus and activated with PDBu (50 ng/ml) and Ionomycin (1 μg/ml) (3 independent infections per experiment). 24 h after activation, cells were dispersed and transferred to new wells and cultured for an additional 3 hours.

**Adhesion assay**

Binding of ICAM-1 complexes to MACS-purified OT-I CD8⁺ T cells. Binding of soluble ICAM-1-Fc-F(ab')₂ complexes were generated by diluting APC-labelled goat anti-human IgG F(ab')₂ fragments (109-135-098, Jackson Immunoresearch) 1:6.25 with recombinant mICAM1-Fc (200 μg/ml final) in HBSS and incubating for 30 min in HBSS at 4 °C. Splenocytes were rested for 3 h in IMDM, 5% FCS at 37 °C, centrifuged

**Fig. 9 | MRTF-dependent F-actin assembly is required for IL-2 delivery. A** Cluster formation requires F-actin. OT-I WT CD8⁺ T cells activated in vitro 24 h on plate-bound anti-CD3/CD28, transferred to uncoated wells and cultured for 3 h with or without 5 μM LatB. Brightfield images are representative of 3 replicates from two independent experiments. **B** OT-I WT CD8⁺ T cells activated as in (**A**), transferred to culture with 5 μM LatB, anti-IL-2, or IL-2, and analysed by immunoblotting. STAT5 Y694 phosphorylation is quantified relative to that in activated cells. **C** Actin expression and F-actin assembly are defective in *Mrtfab⁻/⁻* cells. OT-I WT and OT-I *Mrtfab⁻/⁻* CD8⁺ activated and transferred for 6 h as in (**A**), fixed, stained for intracellular F-actin (phalloidin) and G-actin (DNaseI). One of two independent experiments is shown; datapoints are MFI ± SEM of 4 technical replicates. **D** Defective clustering following 18 h activation by PDBu/ionomycin in vitro. MACS-purified OT-I WT and OT-I *Mrtfab⁻/⁻* CD8+ stimulated for 18 h with PDBu and Ionomycin, fixed, stained with phalloidin, anti-pericentrin antibody and DAPI. Images are maximum intensity projections of Z stacks from representative confocal microscopy images of OT-I WT or OT-I *Mrtfab⁻/⁻* cluster. **E** Cell number, distance to the nearest

neighbour, F-actin content, and sphericity within clusters are shown with mean values ± SEM, unpaired two-tailed t test. The experiment is representative of multiple experiments with similar results. **F** Visualisation of endogenous IL-2. Top, schematic of the catch assay. OT-I WT or OT-I *Mrtfab⁻/⁻* CD8+ were coated with IL-2 catch reagent, stimulated as in (**D**), and IL-2 retained within the cluster revealed using phycoerythrin-conjugated IL-2 antibody. Representative images of cell clusters either stained for IL-2 (upper panels) or false-colored according to IL-2 puncta size (lower panels). Bottom, size distribution of IL-2 puncta (left) and IL-2 signal volumes (right), determined for 8 clusters of each genotype in two independent experiments. Data are shown as mean ± SEM. Scale bar, 10 μm. Unpaired two-tailed *t* test. **G** OT-I WT and OT-I *Mrtfab⁻/⁻* cells were activated as in (**D**), simultaneously infected with lentivirus expressing β-actin or control for 24 h before transfer to new wells for 3 h. Top, representative brightfield images. Bottom, immunoblot analysis of STAT5 Y694 phosphorylation. A representative of two independent experiments is shown. Source data are provided as a Source Data file.

and resuspended in HBSS, 0.5% BSA. Each adhesion reaction (50 μl) contained $20 \times 10^6$ OT-I CD8⁺ T cells/ml, 25 μg/ml ICAM-1 complexes and soluble anti-CD3 (10 μg/ml) and was incubated at 37 °C for the indicated times. OT-I CD8⁺ T cells were stimulated with anti-CD3 (10 μg/ml)). Cells were fixed in 2% PFA for 20 min and binding of ICAM-1 complexes to T cells analysed by flow cytometry.

### Cytolytic assay

For CD8⁺ T effector differentiation, purified naive OT-I CD8⁺ cells were activated on plate-bound anti-CD3/CD28 (10 μg/ml) for 48 h, washed three times, then cultured with 20 ng/ml recombinant IL-2 for an additional 5 days (rmIL-2, Immunotools). CD8 effector cells were mixed with CFSE-stained EL4 target cells, pulsed or not with 1 μg/ml of SIINFEKL peptide, and incubated for 3 h before fixation and staining for intracellular active caspase-3 in EL4 target cells.

### BrdU incorporation

Mice were injected (i.p) either at day 1 or day 6 post-infection with 1 mg of BrdU (BD Biosciences); 48 h later, splenocytes were stained for surface markers and BrdU incorporation according to the manufacturer's protocol.

### RT-qPCR

Total RNA was isolated using the GenElute kit (Sigma) and treated with DNA'ase I. cDNA was synthesised and quantitative PCR performed using SYBR Green Express and a QuantStudio 5 qPCR instrument (Life Technologies). Primers were:

*Actb* intronic: F- CGTAGCGTCTGGTTCCCAAT; R- GTGTGGGCATTTGATGAGCC

*Actb* exonic: F- CGCCACCAGTTCGCCAT; R- CTTTGCACATGCCGGAGC

*Actg* intronic: F- CTGGCCGAGGACATTTTCTG; R- GAAGAAGCCCCGGAATTAGC

*Actg* exonic: F- ATGGAAGAAGAAATCGCCGC; R- AGGGTCAGGATACCCCTCTT

*RpS16*: F- CGCACGCTGCAGTACAAGTTACT; R- ACATGTCCACCACCCTTCACAC

*IL-2* exonic F- TCAGTGCCTAGAAGATGAACTTG; R- TCAAATCCAGAACATGCCGC

*Srf* exonic F- CACCTACCAGGTGTCGGAAT; R- GCTGTGTGGATTGTGGAGGT

### RNAseq and bioinformatics

OT-I WT and OT-I *Mrtfab⁻/⁻* were sorted from live CD8⁺ gated lymph nodes cells derived from tamoxifen-fed bone marrow-reconstituted mice. Each biological replicate represents cells purified from pooled lymph nodes from individual animals. Cells were activated with plate-

bound anti-CD3/CD28 (5 μg/ml) for 24 h, washed and cultured overnight in media supplemented with IL-12 cytokine (20 ng/ml) ("activated/rested" cells). The next day cells were activated with rIL-2 (20 ng/ml) for the times indicated followed by RNA extraction. For RNAseq, RNA was prepared using total RNA GeneElute columns (Sigma). Libraries were prepared using Nugen cDNA synthesis kit.

FASTQ files for each sample were processed via the nf-core/rna-seq pipeline (v3.5)[71] against the Mus Musculus mm10 genome build and RefSeq annotation GCF_000001635.26_GRCm38.p6 using STAR with strandedness set to unstranded. BigWig coverage files were generated using deeptools v3.0.0 with normalisation factors calculated as described[72].

Per gene read counts were retrieved using the R/Bioconductor (v4.0.3) library GenomicAligment (v1.26.0) and the function summarizeOverlaps[73] against the GCF_000001635.26_GRCm38.p6 RefSeq annotation.

All bioinformatics analysis methods and code are available at GitHub repository link https://github.com/fgualdr/maurice_etal_2023_MRTF_SRF_IL2delivery/.

Total and intronic read counts per gene were collected using the summarizeOverlaps method and used for downstream analysis. Sample normalisation was achieved by selecting invariant genes across samples/conditions[42] using the R library GeneralNormalizer. Differentially regulated genes were selected using DESeq2 (R/Bioconductor package version 1.26.0; R version 3.6.2) after turning off the default normalisation that DESeq2 applies. Briefly, dispersion estimates for Negative Binomial distributed data was computed using the DESeq2 function "estimateDispersions" with the fitType variable set to "local" to fit a local regression of log dispersions over log base mean and the individual points weighted by normalised mean count in the local regression. Significance of coefficients in a Negative Binomial GLM was computed using the DESeq2 function "nbinomWaldTest" on the basis of the computed dispersion estimate and a design expressing the counts for each gene as a function of all conditions. Genes differentially expressed considering either all or intronic read counts with an associated absolute Log2FoldChange greater than 1 and an adjusted *p* value less than or equal to 0.05 were collected and clustered according to the z-score across conditions, computed as $\frac{X-\mu}{\sigma}$, where X is the replicate average, and μ and σ are the mean and standard deviation across all conditions, respectively (refer to individual plots and figures). Unsupervised clustering was achieved using dimensionality reduction coupled with Louvain community detection. Gene set enrichment analysis was conducted against the MSigDB (msigdb_v7.5.1) computing the hypergeometric enrichment and the Standardised Jaccard index.

Raw and processed RNAseq data are in GEO, accession code GSE241689.

## Statistical analysis

Data were analysed with Graph Pad Prism 6. Bar and dot charts are expressed as mean ± SEM and data analysed using the unpaired and paired parametric *t* test. *p* values not significant (ns) are indicated.

## Antibodies and staining reagents

Antibodies for extracellular markers were used at a dilution of 1:200 and purchased from: *eBioscience*: CD8 (53-6.7), CD122 (5H4), TCRβ (H57-597), CD44 (IM7), CD5 (53-7.3), CD62L (MEL-14), CCR7 (4B12), NKG2D (CX5), LFA-1/CD11a (M17/4). *BD biosciences*: CD11b (M1/70), CD45.1 (A20), CD45.2 (104), CD4 (RM4-5), CD127 (SB/199), KLRG1 (2F1), TCRβ (H57-597), CD132 (TUGm2), CD25 (PC61), CD69 (H1.2F3). *Biolegend*: CD11a/CD18 (H155-78), CD62L(MEL-14), CD69 (H1.2F3), CCL5 (2E9), B220 (RA3-GB2). Streptavidin Brilliant violet 421*Jackson Immunoresearch*: APC goat anti-human IgG F(ab')₂ fragments (109-135-098), *R&D systems*: rmICAM1-Fc (796-IC-050).

Antibodies for intracellular staining were used at a dilution of 1:200 unless stated otherwise, and purchased from: *eBioscience*: IRF4 (3E4), IFNγ (XMG1.2), Granzyme B (NGZB), Fixable viability dye eFluor780 1:500 (65-0865-14). *BD Biosciences*: Bcl-2 (3F11), Caspase-3 (C92-605), pSTAT5(pY694) (47) (562077), panSTAT5 (89), (610192), TNFα (MP6-XT22), BrdU FITC 1:50 (556028), CD16/32 FcBlock 1:100. *Cell Signalling*: pAKT(T308) (D25E6), pS6 240/244 (D68F8), pS6 235/236 (D57.2.2E), p44/42-ERK (D13.14.4E), pStat5 (pY694) (9351), Stat5 (D206Y) (94205).

*Invitrogen*: anti-DNaseI 488 1:200 (D12371 5 mg/ml), Phalloidin Alexa Fluor 647 1:50 (A22287), CellTrace CFSE 4 μM, SNARF™ 2.8 μM.

Antibodies used for Western blotting were used at 1:1000 unless stated otherwise and purchased from: *Santa-Cruz Biotechnology*: anti-SRF (G20). *BD Biosciences*: panERK (16), panSTAT5 (89). *Bethyl laboratories*: MRTF-B (A302-786A). *Biolegend*: GAPDH 1/5000 W17079A (607901). *Cell Signalling*: pStat5 (pY694) (9351), pan-actin (4968). *Sigma*: β−actin AC-15 1/10000 (A5441). γ−actin (cytoplasmic, lot1108, a gift from Christine Chaponnier and Michael Way). *Abcam: mCherry (ab167453)*.

Antibodies used for in vitro stimulation: *BD Biosciences:* CD28(37.51) (553141), as indicated. *CRUK Facility:* CD3ε (2C11) as indicated.

Blocking/activating antibodies: eBioscience: IL-2 (S4B6) (16-7020-85), control Rat IgG2aκ (eBR2a) (*16-4321-85)*, IL-2 blocking JES6-(1A12) (503706), CD40 1C10 (16-0401-85).

Antibodies and dyes used for Immunofluorescence were purchased from: *Biolegend*: B220-Alexa Fluor-488 1:100 (rat RA3-GB2). *Invitrogen*: Cell Tracker deep red 1 μM (C34565), CellTrace violet 5 μM (C34557). *Abcam*: Pericentrin 1:500 (Ab4448).

Cytokines were purchased from: *ImmunoTools*: rmIL-2 (12340026), *Chiron B.V* rhIL-2 (Proleukin). *R&D*: rmIL-12 (419-ML-050).

## Reporting summary

Further information on research design is available in the Nature Portfolio Reporting Summary linked to this article.

## Data availability

The raw and processed RNAseq data generated in this study have been deposited in the GEO database under accession code GSE241689. RNAseq data readcount and sorted comparison data are provided in Supplementary Data 1 and 2. Source data are provided with this paper.

## Code availability

All bioinformatics analysis methods and code are available at GitHub repository link https://github.com/fgualdr/maurice_etal_2023_MRTF_SRF_IL2delivery/.

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

## Acknowledgements

We thank Adrian Hayday, Caetano Reis e Sousa, Gitta Stockinger, Carola Vinuesa, and lab members for helpful discussions and suggestions during the project and/or comments on the manuscript. We thank Santiago Zelenay for advice, reagents and assistance with the *Listeria* experiments, Sandra Diebold for the anti-DNGR1 397-OVA fusion protein, and Jasmine Abella for the pLVX-mCherry vector. We are grateful to the following Crick Science technology platforms and staff: Phil East and Aengus Stewart at Bioinfomatics and Biostatistics for processing raw RNAseq data; Jimena Perez-Lloret, Jerome Nicod and Advanced Sequencing for library preparation and RNAseq; Sukhveer Purewal and Derek Davies at Flow Cytometry; Camille Charoy at Advanced Light Microscopy for fluorescent image acquisition and analysis; and Clare Watkins and Julie Bee and the Biological Resources Facility for expert support. This work was supported by the Francis Crick Institute, which receives its core funding from Cancer Research UK (CC2102), the UK Medical Research Council (CC2102), and the Wellcome Trust (CC2102). This research was funded in whole, or in part, by the Wellcome Trust CC2102. For the purpose of Open Access, the authors have applied a CC BY public copyright licence to any Author Accepted Manuscript version arising from this submission.

## Author contributions

D.M. constructed the conditional SRF and MRTF mouse strains; conceived and designed the co-transfer-infection protocol and designed, carried out, and interpreted infection, signalling, and imaging experiments; P.C. designed, performed and interpreted experiments; J.D. performed the lentiviral infections and F-actin quantitation assays; F.G. designed and conducted the bioinformatic analyses; B.F. set up the multiphoton live microscopy system and conceived and performed the in vivo imaging. R.T. conceived the project, designed and interpreted experiments, and wrote the paper with D.M.

## Funding

## Competing interests

The authors declare no competing interests.
