## [Peer Review File · Nature Communications]

IL-2 delivery to CD8+ T cells during infection requires MRTF/SRF-dependent gene expression and cytoskeletal dynamicsREVIEWER COMMENTS

Reviewer #1 (Remarks to the Author):

The manuscript by Maurice et al. investigates the functional requirement of serum response factor (SRF) and its cofactors in CD8+ T cells that play a critical role in infection both initially in pathogen clearance and then subsequently as memory cells. Different signals are needed to drive proliferation during distinct stages of immune response, but the mechanisms in terms of transcriptional regulation of these processes are not fully understood. SRF is an essential transcription factor that is controlled by two signal-regulated transcription cofactor families, the TCFs and MRTFs. TCFs are regulated by ERK signaling and the TCF-SRF complex regulate the expression of many genes involved with cell proliferation. MRTFs, on the other hand, are regulated by actin dynamics, and MRTF/SRF complex regulates the expression of especially cytoskeletal genes. Hence generally, TCFs, rather than MRTFs, are implicated in proliferative responses.

In this manuscript, the authors use several elegant assays to demonstrate that MRTF/SRF are dispensable for the initial TCR-mediated CD8+ T cell proliferation but required for the subsequent IL-2 dependent proliferation. The authors show that MRTF-null cells display reduced response to IL-2 signaling, which they link to defective cytoskeletal gene expression and thereby e.g. reduced cell clustering.

Overall, this a carefully conducted study and the functional requirement for MRTFs towards cell proliferation via cytoskeletal gene expression is very interesting.

Major concerns

If the manuscript is considered for Nature communications, the writing style should be significantly simplified. In its present state, the manuscript is very "immunological" for general audience.

Can the cytoskeletal defect in MRTF null cells be rescued to prove that the proliferation defect is specifically due to transcription of cytoskeletal genes? For example, it has previously been shown in both human breast cancer cells depleted for MRTFs and in mal-d mutant Drosophila that re-expression of actin rescues, at least part of, the phenotypic consequences of Mal/MRTF loss (invasive migration and border cell migration, respectively). 10.1101/gad.237743.114

To assess the role of TCFs, the authors test Elk4 null cells. However, their own previous results have shown that at least in the thymus, Elk4 and Elk1 act redundantly to limit the generation of innate-like CD8+ T cells. Is it possible that Elk1 plays a role here also?

In the RNA-seq data, what happens to the TCF target genes in MRTF KO cells? Presumably they should be largely upregulated due to the competition between MRTFs and TCFs.

Minor points

In figure 1A, the schematic of the experimental set up: (day 1) under the "Purify CD8..." is confusing, because the next part mentions day 0. Presumably it is meant here that this part took 1 day?

First reference to figure 1D is missing from the text.

In figure 4A, what is the difference between right and left panels?

Please check the figure legend/panels for figure 8; there is some confusion.

Reviewer #2 (Remarks to the Author):

This article by Maurice et al. entitled "Defective cytoskeletal dynamics underlies the essential role of MRTF-SRF in IL-2 delivery to CD8+ T cells during infectious challenge" reports a solid study that convincingly shows the requirement of MRTF-SRF activity for IL-2 responsiveness during LM-OVA infection. Using well-calibrated in vivo experiments and Srf or Mrtfa/b KO mice models, the authors provide evidence that MRTF-SRF activity favours actin expression and polymerisation, and cytoskeletal modifications required for migration, homotypic T-T clusters formation, IL-2 responsiveness, and consequently, cellular expansion. Based on these profound cytoskeletal blocks, it is surprising that initial T cell activation events and early proliferation are not affected in Srf or Mrtfa/b KO mice models. It is a particularly interesting and thorough study. The experiments are precisely conducted and the results are convincing.

Minor comments:

The authors have conducted a RNAseq study that has identified actin and cytoskeletal (Wdr1, Myo9) genes that are down-regulated in KO cells. It would be useful to know if alterations in the expression of these genes is indeed the cause of the observed phenotype. Does the knocking-down of these genes in WT CD8+ T cells elicit inhibition of T-T clustering and IL-2 responsiveness?

Reviewer #3 (Remarks to the Author):

The manuscript by Maurice et al presents an array of clear and concise data, including whole animal infection in the settings of radiation bone marrow chimeras and adoptive cellular transfer recipients, intracellular proetin staining, gene expression profiling, and multiphoton confocal microscopy, in support of the finding that the transcription factor SRF, working through MRTF co-factors, controls paracrine secretion of IL-2 in CD8+ T cells and all of the downstream sequelae of IL-2 signals transmitted from membrane receptor to nuclear gene expression and functional development of committed effector cells. While there is certainly value for the field to have these results clearly demonstrated and reported, the question of whether this is truly an IL-2 specific defect or, in light of known roles of SRF and the MRTFs in regulating cytoskeletal dynamics, this reflects a more global defect in extracellular export that might be predicted by their absence. As the authors have already demonstrated that activation (TCR)-mediated production of the effector cytokine TNF and IFN-g are unaffected in SRF and MRTF-deficient CD8+ T cells when measured intracellularly, it would seem that comparing the secretion of these by ELISA with wildtype cells could help to address this important issue.

1. SUMMARY

We thank the three reviewers for their generally positive view of our paper, which we have now revised to take into account their specific comments in the following way:

- ELISA assays now demonstrate no general defect in secretion in *Srf-* or *Mrtfab*-null cells (Fig 4E, 4F, new S4F, text pp10-11)
- Cluster formation and STAT5 Y694 phosphorylation now shown to require F-actin (new Fig 9A, new 9B, new S9D; text p17)
- Levels of total actin protein and F-actin assembly shown to be greatly reduced in *Mrtfab*-null cells (new Fig 8F, new Fig 9C; new Fig S9A text pp17-18).
- Exogenous expression of β -actin shown to restore cluster formation in *Mrtfab*^{-/-} cells, showing that β -actin expression can compensate for loss of the MRTFs (new Fig 9G; text in abstract and pp19, 24-25)
- Role of TCF-SRF signalling now discussed in more detail in Results p9 and Discussion p20-21
- Exogenous actin expression experiments by Salvany et al now cited and discussed Intro p4, Results p18, Discussion p24, new Fig S9G.

A detailed inventory of changes to Figures, and responses to the individual referees are summarised below.

2. CHANGES TO FIGURES (New data panels are highlighted in yellow)

- Figure 1 panel A: headings corrected
- Figure S1 no change
- Figure 2 panel A: headings corrected
- Figure S2 no change
- Figure 3 panels A,D: headings corrected
- Figure S3 no change
- Figure 4 Old Fig 4D,4E dotplots merged to become new Fig 4D
Old Fig 4D,4E CFSE profiles moved to Fig S4
New Fig 4E,4F show IL-2 transcription and ELISA (were old Fig 7A,7B).
- Figure S4 new Fig S4D, S4E from old Fig 4D,4E;
new panel S4E was old Fig S4C;
new Fig S4F presents new TNF α and IFN γ cytokine ELISA data.
- Figure 5 no change
- Figure S5 no change
- Figure 6 no change
- Figure S6 new Fig S6A moved from old S7C
Fig S6B is old Fig S6A
- Figure 7 new Fig 7D,7E show defective clustering in vivo (were old Fig 9D,9E);
lower panels of Fig 7C become new Fig S7C

- Figure S7 panels S7A,S7B moved to new Fig 4E,4F;
panels S7C,S7D relabelled as S7A, S7B;
new panel S7C moved in from old Fig 7C;
panels S7D, S7E relabelled as S7D, S7E
new panels S7F, S7G moved in from old Fig S9D, S9E
- Figure 8 **New Fig 8F** now shows that levels of total actin protein and β - and γ -actin are MRTF-dependent in CD8⁺ T cells, and that they increase upon activation by TCR crosslinking and transfer.
- Figure S8 no change
- Figure 9 Old Fig 9D,9E (defective in vivo clustering) moved to new Fig 7D,7E.
New Fig 9A: clustering in wildtype cells activated by TCR crosslinking is abolished when actin polymerisation is inhibited by Latrunculin B.
New Fig 9B: STAT5 Y694 phosphorylation in wildtype cells is reduced when actin polymerisation is inhibited by Latrunculin B.
New Fig 9C: FACS analysis showing that both basal and activation-induced F- and G-actin levels are substantially reduced in *Mrtfab*^{-/-} cells (see also new Fig.S9A)
New Fig 9G: exogenous expression of β -actin restores cluster formation in *Mrtfab*^{-/-} cells, rendering cluster formation essentially independent of the MRTFs, and also potentiates STAT5 Y694 phosphorylation.
- Figure S9 **New Fig S9A**: F-actin sedimentation assay showing basal and activation-induced F- and G-actin levels are substantially reduced in *Mrtfab*^{-/-} cells.
New Fig S9D: clustering in wildtype cells activated by PDBu / ionomycin is abolished when actin polymerisation is inhibited by Latrunculin B.
New Fig S9F: exogenous expression of β -actin restores cluster formation in *Mrtfab*^{-/-} cells (20x magnification).
New Fig S9G: MRTF-SRF transcriptional targets and cytoskeletal regulatory interactions.

3. REVIEWERS COMMENTS (our detailed responses in blue)

Reviewer #1 (Remarks to the Author):

The manuscript by Maurice et al. investigates the functional requirement of serum response factor (SRF) and its cofactors in CD8+ T cells that play a critical role in infection both initially in pathogen clearance and then subsequently as memory cells. Different signals are needed to drive proliferation during distinct stages of immune response, but the mechanisms in terms of transcriptional regulation of these processes are not fully understood. SRF is an essential transcription factor that is controlled by two signal-regulated transcription cofactor families, the TCFs and MRTFs. TCFs are regulated by ERK signaling and the TCF-SRF complex regulate the expression of many genes involved with cell proliferation. MRTFs, on the other hand, are regulated by actin dynamics, and MRTF/SRF complex regulates the expression of especially cytoskeletal genes. Hence generally, TCFs, rather than MRTFs, are implicated in proliferative responses.

In this manuscript, the authors use several elegant assays to demonstrate that MRTF/SRF are dispensable for the initial TCR-mediated CD8+ T cell proliferation but required for the subsequent IL-2 dependent proliferation. The authors show that MRTF-null cells display reduced response to IL-2 signaling, which they link to defective cytoskeletal gene expression and thereby e.g. reduced cell clustering.

Overall, this a carefully conducted study and the functional requirement for MRTFs towards cell proliferation via cytoskeletal gene expression is very interesting.

Major concerns

If the manuscript is considered for Nature communications, the writing style should be significantly simplified. In its present state, the manuscript is very “immunological” for general audience.

In our original manuscript we tried to explain the systems used wherever possible indeed specialist immunologist readers considered some of this unnecessary. We have now done our best to replace specialist terms and abbreviation with explanatory phrases wherever possible to help non-immunologist reader.

Text amended p3 lines 3-4; p6 lines 82-83; p7, lines 94-95; p8, lines 121-122 p9, line 150; p12 lines 218-219.

Can the cytoskeletal defect in MRTF null cells be rescued to prove that the proliferation defect is specifically due to transcription of cytoskeletal genes? For example, it has previously been shown in both human breast cancer cells depleted for MRTFs and in mal-d mutant Drosophila that re-expression of actin rescues, at least part of, the phenotypic consequences of Mal/MRTF loss (invasive migration and border cell migration, respectively). 10.1101/gad.237743.114.

Our transcriptional analysis indicates that KO cells exhibit defective expression of many genes involved in actin cytoskeletal structures and regulation. We have focussed on the role of the cytoskeletal actins.

We now show that clustering is critically dependent on F-actin assembly, and that in cells exogenously expressing actin, cluster formation is no longer dependent on MRTF activity. Specifically:

- Expression of both β - and γ -actin is deficient in MRTF dKO cells, and that CD8 T cell activation leads to increased levels of total actin and individual β - and γ -actins (new Fig. 8F).
- Clustering of wildtype activated CD8+ T cells following TCR activation and transfer to is disrupted when F-actin assembly is inhibited using with Latrunculin B, which prevents actin treadmilling, indicating its dependence on F-actin integrity (new Fig. 9A).
- STAT5 Y694 phosphorylation in clustered wildtype CD8+ T cells is reduced upon LatB treatment (new Fig. 9B)
- Activation of CD8+ T cells induces F-actin assembly, as judged by FACS assays for F- and G-actin with Phalloidin and DNase I respectively (new Fig. 9B) or by F-actin pelleting assays (new Fig. S9A).
- F-actin levels are substantially reduced in MRTF-null cells (new Fig. 9B, Fig. S9A).
- While inactivation of MRTF abolishes clustering following activation, exogenous expression of β -actin both substantially restores clustering (new Fig. 9G) and potentiates STAT5 Y694 phosphorylation.
- Abstract revised; Introduction amended, p4 lines 51-52; Results new text p17 lines 332-335, p17 lines 344-351; pp24/25 lines 373-382; Discussion p20 lines 392-398, p24 lines 493-507. Schematic summary, new Fig. S9G.

Salvany et al (2014) is now cited Intro p4 lines 51-52; Results p18 lines 373-374; Discussion p24 lines 491-498; Fig. S9G legend.

To assess the role of TCFs, the authors test Elk4 null cells. However, their own previous results have shown that at least in the thymus, Elk4 and Elk1 act redundantly to limit the generation of innate-like CD8+ T cells. Is it possible that Elk1 plays a role here also?

We find that the SRF-null proliferative phenotype recapitulates the MRTF-null phenotype. This shows that even with the TCF/SRF arm is still intact, TCF-SRF signalling cannot compensate for the loss of the MRTFs. It also renders it very unlikely that the residual proliferation seen in MRTF-null cells is mediated by TCF signalling.

It remains possible that TCF-SRF signalling does contribute to other aspects of the infectious response, in CD8+ or other immune cells, and inactivation of both Elk4/SAP-1 and other TCFs may accentuate such phenotypes. However, for logistical reasons we have not pursued this agenda instead focussing specifically on the MRTF-SRF arm of the network.

Issue addressed in Results, page p9 lines 158-162; Discussion, p20 lines 399-407.

In the RNA-seq data, what happens to the TCF target genes in MRTF KO cells? Presumably they should be largely upregulated due to the competition between MRTFs and TCFs.

Like the referee, we considered this a possibility; however, examination of the data shows it not to be the case. Competition will reflect the relative abundance of the cofactors, and their subcellular location. A fundamental difference between the TCFs and the MRTFs is that the TCFs are constitutively nuclear whereas the inactive MRTFs are predominantly cytoplasmic. Thus, while removal of TCFs will reveal the basal activity of MRTFs, removal of the MRTFs is unlikely to potentiate TCF activity.

Issue addressed in Discussion, p20-p21 lines 407-411.

Minor points

In figure 1A, the schematic of the experimental set up: (day 1) under the “Purify CD8...” is confusing, because the next part mentions day 0. Presumably it is meant here that this part took 1 day?

No, this was a typo, sorry! It should have been -1 ie the day before infection which is taken as time day zero in the infection timecourse in Figure 1C.

Captions corrected. Similar error corrected in Figure 2A, 3A, 3D.

First reference to figure 1D is missing from the text.

Sorry! Text reordered on p7 lines 98-99.

In figure 4A, what is the difference between right and left panels?

These are two experiments, first to capture the identical kinetics of the initial proliferation, the second to capture the onset of the defect. Revised Figure 4 legend now makes this clear.

Please check the figure legend/panels for figure 8; there is some confusion.

Drafting error, legend corrected. (new Figure 8F immunoblot now shows total actin levels as well as those of β - and γ -actins).

Reviewer #2 (Remarks to the Author):

This article by Maurice et al. entitled "Defective cytoskeletal dynamics underlies the essential role of MRTF-SRF in IL-2 delivery to CD8+ T cells during infectious challenge" reports a solid study that convincingly shows the requirement of MRTF-SRF activity for IL-2 responsiveness during LM-OVA infection. Using well-calibrated in vivo experiments and Srf or Mrtfa/b KO mice models, the authors provide evidence that MRTF-SRF activity favours actin expression and polymerisation, and cytoskeletal modifications required for migration, homotypic T-T clusters formation, IL-2 responsiveness, and consequently, cellular expansion. Based on these profound cytoskeletal blocks, it is surprising that initial T cell activation events and early proliferation are not affected in Srf or Mrtfa/b KO mice models. It is a particularly interesting and thorough study. The experiments are precisely conducted and the results are convincing.

Minor comments:

The authors have conducted a RNAseq study that has identified actin and

cytoskeletal (*Wdr1*, *Myo9*) genes that are down-regulated in MRTF dKO cells. It would be useful to know if alterations in the expression of these genes is indeed the cause of the observed phenotype. Does the knocking-down of these genes in WT CD8⁺ T cells elicit inhibition of T-T clustering and IL-2 responsiveness?

Our transcriptional analysis indicates that KO cells exhibit defective expression of many genes involved in actin cytoskeletal structures and regulation. We have focussed on the role of the cytoskeletal actins.

It will certainly be of interest to examine the contributions of individual MRTF-SRF target genes to the response to LM-OVA infection. However, we have instead extended our previous findings by investigating the role of F-actin in cluster formation.

We now show that clustering is critically dependent on F-actin assembly. Salvany et al (2014) previously showed that in some contexts MRTF phenotypes can be suppressed by exogenous actin expression, and we show that in *Mrtfab*^{-/-} cells exogenously expressing actin, cluster formation is substantially restored. Exogenous actin expression also effectively renders clustering no longer dependent on MRTF activity. Specifically:

- Expression of both β - and γ -actin is deficient in MRTF dKO cells, and that CD8 T cell activation leads to increased levels of total actin and individual β - and γ -actins (new Fig. 8F).
- Clustering of wildtype activated CD8⁺ T cells following TCR activation and transfer to is disrupted when F-actin assembly is inhibited using with Latrunculin B, which prevents actin treadmilling, indicating its dependence on F-actin integrity (new Fig. 9A).
- STAT5 Y694 phosphorylation in clustered wildtype CD8⁺ T cells is reduced upon LatB treatment (new Fig. 9B)
- Activation of CD8⁺ T cells induces F-actin assembly, as judged by FACS assays for F- and G-actin with Phalloidin and DNase I respectively (new Fig. 9B) or by F-actin pelleting assays (new Fig. S9A).
- F-actin levels are substantially reduced in MRTF-null cells (new Fig. 9B, Fig. S9A).
- While inactivation of MRTF abolishes clustering following activation, exogenous expression of β -actin both substantially restores clustering (new Fig. 9G) and potentiates STAT5 Y694 phosphorylation.
- Abstract revised; Introduction amended, p4 lines 51-52; Results new text p17 lines 332-335, p17 lines 344-351; pp24/25 lines 373-382; Discussion p20 lines 392-398, p24 lines 493-507. Schematic summary, new Fig. S9G.
- Salvany et al (2014) in which actin overexpression rescues MRTF-null phenotypes, now cited Intro p4 lines 51-52; Results p18 lines 373-374; Discussion p24 lines 491-498; Fig. S9G legend.

Reviewer #3 (Remarks to the Author):

The manuscript by Maurice et al presents an array of clear and concise data, including whole animal infection in the settings of radiation bone marrow chimeras and adoptive cellular transfer recipients, intracellular protein staining, gene expression profiling, and multiphoton confocal microscopy, in support of the finding that the transcription factor SRF, working through MRTF co-factors, controls paracrine secretion of IL-2 in CD8+ T cells and all of the downstream sequelae of IL-2 signals transmitted from membrane receptor to nuclear gene expression and functional development of committed effector cells.

While there is certainly value for the field to have these results clearly demonstrated and reported, the question of whether this is truly an IL-2 specific defect or, in light of known roles of SRF and the MRTFs in regulating cytoskeletal dynamics, this reflects a more global defect in extracellular export that might be predicted by their absence.

As the authors have already demonstrated that activation (TCR)-mediated production of the effector cytokine TNF and IFN- γ are unaffected in SRF and MRTF-deficient CD8+ T cells when measured intracellularly, it would seem that comparing the secretion of these by ELISA with wildtype cells could help to address this important issue.

Thank you for your positive comments on our work. You raise the question of whether the effect on IL-2 signalling reflects a more general failure of secretion in OT-I MRTF-null cells, since only intracellular staining was shown for TNF α and IFN γ in Figure 1.

We thank you for bringing the secretion issue to our attention, as it made us realise we underemphasised whole issue of cytokine secretion following activation in the original manuscript.

In Figure 4/S4 we showed that activation of CD8 T cells results in normal initial proliferation. We have now extended this to show that IL-2 and other cytokine expression and secretion is induced normally following activation of MRTF-null cells.

We show that transcriptional activation and secretion of IL-2 occurs normally following TCR activation of SRF- and MRTF-null cells (new Fig 4E, 4F), and that secretion of TNF α and IFN γ also occurs normally following activation of CD8+ cells by PDBu / ionomycin (new Fig. S4F).

These results support our proposal that MRTF-SRF signalling is needed for IL-2 cytokine presentation to neighbouring cells by cell contact rather than cytokine secretion.

- Fig 4D – CFSE plots moved to supplementary Fig S4C, S4D.
- Fig 4E, 4F – data moved from old Fig S7A, S7B.
- New Figure S4F.
- Text amended, p10 lines 182-186; Discussion p20 line 393, p22 lines 452-454.

REVIEWERS' COMMENTS

Reviewer #1 (Remarks to the Author):

I am satisfied the revisions. The rescue of some of the phenotypes with actin expression is elegant, and further emphasizes the evolutionarily conserved role of MRTFs in actin gene expression.

Reviewer #2 (Remarks to the Author):

The authors have addressed my comments satisfactorily.

Reviewers' comments

Reviewer #1 (Remarks to the Author):

I am satisfied the revisions. The rescue of some of the phenotypes with actin expression is elegant, and further emphasizes the evolutionarily conserved role of MRTFs in actin gene expression.

Reviewer #2 (Remarks to the Author):

The authors have addressed my comments satisfactorily

Authors response

We thank the reviewers for their positive response and look forward to publication.